# The RhoB p.S73F mutation leads to cerebral palsy through dysregulation of lipid homeostasis

Xinyu Wu [1,9], Ruonan Liu [1,9], Zhongtian Zhang [2,3,4,5,6,7], Jie Yang [1], Xin Liu [1], Liqiang Jiang [1], Mengmeng Fang [1], Shoutang Wang [8], Liangxue Lai [1,2,3,4,5,6,7✉], Yuning Song [1✉] & Zhanjun Li [1✉]

## Abstract

**Cerebral palsy (CP) is a prevalent neurological disorder that imposes a significant burden on children, families, and society worldwide. Recently, the RhoB p.S73F mutation was identified as a de novo mutation associated with CP. However, the mechanism by which the RhoB p.S73F mutation causes CP is currently unclear. In this study, rabbit models were generated to mimic the human RhoB p.S73F mutation using the SpG-BE4max system, and exhibited the typical symptoms of human CP, such as periventricular leukomalacia and spastic-dystonic diplegia. Further investigation revealed that the RhoB p.S73F mutation could activate ACAT1 through the LYN pathway, and the subsequently altered lipid levels may lead to neuronal and white matter damage resulting in the development of CP. This study presented the first mammalian model of genetic CP that accurately replicates the RhoB p.S73F mutation in humans, provided further insights between RhoB and lipid metabolism, and novel therapeutic targets for human CP.**

**Keywords** Cerebral Palsy; RhoB; Rabbit Model; Neuromotor Impairment; Lipid Metabolism
**Subject Categories** Genetics, Gene Therapy & Genetic Disease; Neuroscience

## Introduction

Cerebral palsy (CP) is a condition characterized by nonprogressive symptoms in the developing fetal or newborn brain, leading to lifelong impairments in movement and posture (Rosenbaum, Paneth et al, 2007), which imposes a significant burden on children, families, and society worldwide (Yang, Xia et al, 2021). CP describes the most common physical disability with other neurodevelopmental and neurofunctional disorders in childhood, and the diagnosis could be made before 6 months (Novak, Morgan et al, 2017).

Neonatal asphyxia was initially identified as a major cause of CP (Clark, Ghulmiyyah et al, 2008; Little and W., 1966; Nelson and Karin, 2008). However, only 14.5% of CP cases are actually caused by birth asphyxia (Graham, Ruis et al, 2008). Recent advancements in clinical practice, such as electronic fetal monitoring (EFM) and other technologies, have significantly enhanced obstetrics and infant care of CP, while the global incidence of CP remains constant at two to three cases per 1000 live births (Clark and Hankins, 2003). Increasing evidence indicates that CP, along with other developmental brain disorders such as intellectual disability and autism spectrum disorders, may also be associated with various rare genetic abnormalities (Choong Yi Fong et al, 2010; Li, Zhou et al, 2021; Montenegro, Cendes et al, 2005).

Recently, two separate cases revealed for the first time that the RhoB p.S73F mutation mediates CP (Jin, Lewis et al, 2020). In addition, the somatic mutations of RhoB are typically linked to cancer cell migration and invasion (Chen, Sun et al, 2000; Sjöblom, Jones et al, 2006). However, the exact mechanism of the RhoB gene in neurological disorders remains unclear.

Most of the CP animal models for the current research are simulated by toxins or surgical injury to the brain (Cavarsan, Gorassini et al, 2019; Visco, Toscano et al, 2021). However, only a few of these models show diagnostic signs of spasticity and motor deficits (Brandenburg, Fogarty et al, 2019). For instance, in a hamster model of brain malformation induced with glutamate analogs, neurons migrate abnormally, but no motor symptoms are observed (Marret, Gressens et al, 1996). Similarly, mice (Zaghloul, Patel et al, 2017) or rats (Durán-Carabali, Sanches et al, 2017) asphyxiated will exhibit relatively mild motor symptoms that eventually normalized over time (Brandenburg et al, 2019). Animal models of in-utero injury could exhibit brain abnormalities; however, due to feeding disorders, these models often do not survive long enough to manifest motor symptoms (Myers, 1975). In addition, some of the gene-edited models showed a spasticity

[1]State Key Laboratory for Diagnosis and Treatment of Severe Zoonotic Infectious Diseases, Key Laboratory for Zoonosis Research of the Ministry of Education, and College of Veterinary Medicine, Jilin University, Changchun 130062, China. [2]CAS Key Laboratory of Regenerative Biology, Guangdong Provincial Key Laboratory of Stem Cell and Regenerative Medicine, Guangzhou Institutes of Biomedicine and Health, Chinese Academy of Sciences, Guangzhou 510530, China. [3]Sanya Institute of Swine Resource, Hainan Provincial Research Centre of Laboratory Animals, Sanya 572000, China. [4]Guangdong Provincial Key Laboratory of Large Animal Models for Biomedicine, Wuyi University, Jiangmen 529020, China. [5]Jilin Provincial Key Laboratory of Animal Embryo Engineering, Key Laboratory of Zoonosis Research, Ministry of Education, College of Veterinary Medicine, Jilin University, Changchun 130062, China. [6]Institute of Stem Cells and Regeneration, Chinese Academy of Sciences, Beijing 100039, China. [7]Research Unit of Generation of Large Animal Disease Models, Chinese Academy of Medical Sciences, Guangzhou 510530, China. [8]School of Biomedical Sciences, LKS Faculty of Medicine, The University of Hong Kong, Pokfulam, Hong Kong. [9]These authors contributed equally: Xinyu Wu, Ruonan Liu. ✉E-mail: lai_liangxue@gibh.ac.cn; songyuning0313@jlu.edu.cn; lizj_1998@jlu.edu.cn

phenotype similar to CP, such as the SPA mouse model, but the specific mutation responsible for the phenotype has not been associated with any known cases of genetic CP (Brandenburg, Gransee et al, 2018; Cosgrove and Graham, 1994; Feng, Tintrup et al, 1998; Huebner, Stein et al, 2001). Thus, there is currently no suitable mammalian model of CP available for study.

Animal models are essential for the understanding of brain development and neurological disorders (Muñoz-Moreno, Arbat-Plana et al, 2013). In neuroscience research, rats, mice, rabbits, pigs, and non-human primates are commonly utilized. Compared with rats and mice, rabbits exhibit perinatal brain development characteristics similar to those of humans (Binderman, Harel et al, 1988), particularly in terms of white matter maturation (Derrick, Drobyshevsky et al, 2007). The structural damage to white matter observed in hypoxic-ischemic rabbits closely mirrors that found in human clinical cases (Drobyshevsky, Jiang et al, 2014). Additionally, rabbits are relatively easy to care for and breed in comparison to pigs and non-human primates. Thus, the rabbit model of CP is emerging as a practical model for studying movement disorders (Cavarsan et al, 2019).

In this study, the rabbit model of CP successfully replicated key features of motor impairment observed in human CP cases, including a forward tilt of the trunk, hypertonia of the lower limbs, and a scissor-like gait. Moreover, based on the rabbit model of CP, Further investigation revealed that the RhoB p.S73F mutation could activate ACAT1 through the LYN pathway and result in the development of CP. In conclusion, the first mammalian model of genetic CP disease was successfully generated and revealed the mechanism by which the RhoB p.S73F mutation could disrupt lipid homeostasis, resulting in brain damage, providing a new perspective for the prevention and treatment of hereditary CP.

# Results

## RhoB$^{S73F/S73F}$ rabbits were successfully generated by the SpG-BE4max system

The RhoB p.S73F mutation is located in the Switch II structural domain, which is highly conserved across species (Fig. 1A). To create a rabbit model that replicates the RhoB p.S73F mutation in human CP, a sgRNA was designed (Fig. 1B). Cas9 mRNA and sgRNA were complexed and injected into embryos at the one-cell stage, then the editing efficiency was examined at the blastocyst stage. The results revealed that 25/30 of embryos (83.3%) carried the RhoB p.S73F mutation (Fig. EV1A), and homozygous mutations were determined in 3/30 of embryos (10%). To generate RhoB p.S73F rabbits, a total of 120 injected zygotes were transferred to four surrogate rabbits, and two surrogate rabbits gave birth to 11 live pups (Fig. EV1B,C). 3/11 newborn pups (27.3%) carried the homozygous RhoB p.S73F mutation (RhoB$^{S73F/S73F}$ rabbits), while 4/11 pups (36.4%) carried the heterozygous mutation (RhoB$^{S73F/+}$ rabbits). The stable inheritance of the RhoB mutation was obtained in F1 generation rabbits (Fig. EV1D).

## Significant neuromotor impairment was observed in RhoB$^{S73F/S73F}$ rabbits

The two human patients diagnosed with RhoB-mutant CP both carried the RhoB p.F73S heterozygous mutation and exhibited spastic-dystonic movement disorder (Jin et al, 2020). In this study, a relatively

lower mortality rate (50% within 30 weeks, Fig. EV1E,F) and less severe neuromotor impairment (Fig. EV1G–I; Movies EV1–3) were observed in RhoB$^{S73F/+}$ rabbits. While, the significantly reduced body weight from 4 weeks after birth and death within 26 weeks were observed in RhoB$^{S73F/S73F}$ rabbits compared to age-matched wild-type (WT) controls (Fig. 1C,D). The severe motor and postural control issues, such as a forward tilt of the trunk, hypertonia of the lower limbs, a scissor-like gait (Fig. 1E; Movie EV4–7), and the reduced stride length, step width, and step length (Fig. 1F,G) were determined in 2-month-old RhoB$^{S73F/S73F}$ rabbits. Thus, the homozygous mutation of RhoB$^{S73F/S73F}$ rabbits, which exhibited a more severe phenotype of CP, were used for the following study.

Additionally, significantly reduced motility using the open field test (Fig. 1H), a longer rollover time using the rollover reflex test (Fig. 1I), and decreased flexion strength (Figs. 1J and EV2A) were observed in RhoB$^{S73F/S73F}$ rabbits than in WT controls.

Notably, RhoB$^{S73F/S73F}$ rabbits were unable to locate hidden nuts in cotton within 1 h (Figs. 1K and EV2B), which is consistent with the previous study, showed sensory processing difficulties and cognitive impairment in individuals with CP (Morgan and McGinley, 2018; Stadskleiv, 2020). Those results showed that significant neuromotor impairment was observed in RhoB$^{S73F/S73F}$ rabbits.

## Hind limb neurogenic lesions were observed in RhoB$^{S73F/S73F}$ rabbits

Impaired mobility in patients with CP is often influenced by spasticity and dysregulation of the musculoskeletal system (Morgan and McGinley, 2018). In this study, asymmetry in the L6 and L7 vertebral spaces, lumbar-sacral scoliosis, and muscle atrophy was determined in RhoB$^{S73F/S73F}$ rabbits by using x-ray Radiographs (Fig. 2A). The significantly increased serum creatine kinase (CK) (Fig. 2B), decreased mean fiber area and fiber type grouping (Fig. 2C,D) were observed in 12-week-old RhoB$^{S73F/S73F}$ rabbits than in WT controls. Thus, typical signs of muscle atrophy were observed in gastrocnemius muscle sections of RhoB$^{S73F/S73F}$ rabbits.

Then, needle electromyography (EMG) was used to test the critical muscle parameters, including the insertion potentials (Ips, potential analysis during needle movement), resting potentials (RPs, potential analysis with the muscle at rest), and motor unit potentials (MUPs, potential analysis at slight voluntary muscle contraction). The results showed the significant prolongation of the IPs (Fig. EV2C) and high-frequency discharges of the RPs (Fig. 2E), and reduced wave amplitude of the MUPs (Fig. 2F) in hind limb gastrocnemius muscle of RhoB$^{S73F/S73F}$ rabbits. Furthermore, motor nerve conduction velocity (MNCV, velocity of electrical impulses through nerves) tests revealed longer latencies, reduced wave amplitudes, and a significant slowing of MNCV in the hind limb of RhoB$^{S73F/S73F}$ rabbits (Fig. 2G). These findings indicate neurogenic pathology in the skeletal muscle of RhoB$^{S73F/S73F}$ rabbits.

## Brain tissue damage was observed in RhoB$^{S73F/S73F}$ rabbits

To observe brain lesions, magnetic resonance imaging (MRI) scans were conducted on the brains of 12-week-old rabbits. The defect in the corpus callosum, enlargement of the lateral ventricles, softening of the paraventricular white matter, and enlargement of the third ventricle were observed in RhoB$^{S73F/S73F}$ rabbits compared to WT

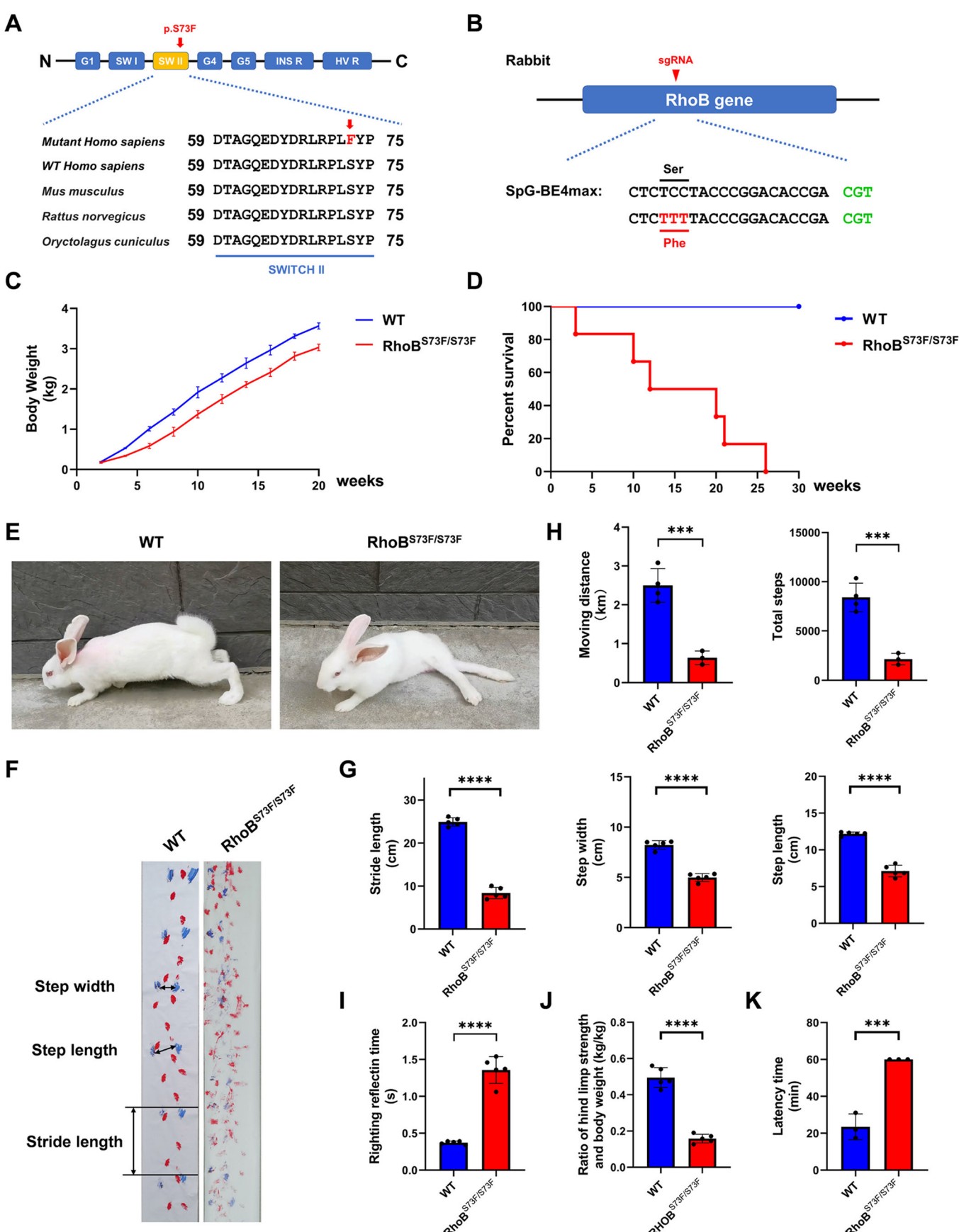

Figure 1.  Significant neuromotor impairment was observed in RhoB^S73F/S73F rabbits.

(A) Multiple species sequence alignment of the SWITCH II domain of RhoB protein. The mutant residue is shown in red. Protein data were obtained from the NCBI Protein (https://www.ncbi.nlm.nih.gov/protein). (B) Schematic diagram of the design location of rabbit RhoB gene SgRNA. Mutant bases are shown in red. (C) Body weight curves of age-matched WT controls ($n = 3$) and RhoB^S73F/S73F rabbits ($n = 3$). A significantly reduced body weight was observed in RhoB^S73F/S73F rabbits. (D) Survival curves of age-matched WT controls ($n = 4$) and RhoB^S73F/S73F rabbits ($n = 6$). A significantly reduced survival rate was observed in RhoB^S73F/S73F rabbits. (E) Representative postural images of 2-month-old WT controls ($n = 3$) and RhoB^S73F/S73F rabbits ($n = 3$). Severe motor and postural control issues were observed in RhoB^S73F/S73F rabbits. (F, G) Representative gait trajectories (F) and gait analysis (G) of 2-month-old WT controls ($n = 5$) and RhoB^S73F/S73F rabbits ($n = 5$). The significantly varied pace trajectory was observed in RhoB^S73F/S73F rabbits. Red indicates front paw tracks; blue indicates hind paw tracks. The dataset includes measurements of stride length, step width, and step length of the hind limb footprint of rabbits during movement. See details on $P$ values in Appendix Table S1. (H) Open field detection of the moving distance and total steps in 2-month-old WT controls ($n = 4$) and RhoB^S73F/S73F rabbits ($n = 3$) during 8 h. The significantly reduced motility was observed in RhoB^S73F/S73F rabbits. See details on $P$ values in Appendix Table S2. (I) Righting reflex detection of 2-month-old WT controls ($n = 5$) and RhoB^S73F/S73F rabbits ($n = 5$). A significantly longer rollover time was observed in RhoB^S73F/S73F rabbits. See details on $P$ values in Appendix Table S3. (J) Muscle tone detection of the lower limbs in 2-month-old WT controls ($n = 5$) and RhoB^S73F/S73F rabbits ($n = 5$). A significantly weaker muscle tone was observed in RhoB^S73F/S73F rabbits. See details on $P$ values in Appendix Table S4. (K) Buried nut trials of 2-month-old WT controls ($n = 3$) and RhoB^S73F/S73F rabbits ($n = 3$). The RhoB^S73F/S73F rabbits were unable to locate hidden nuts in cotton within the given 1-h time frame. See details on $P$ values in Appendix Table S5. Data information: Data represent different numbers ($n$) of biological replicates. In (G–K), data were presented as mean ± SD (Unpaired two-tailed Student's $t$-tests). ***$P \leq 0.001$, ****$P \leq 0.0001$. Source data are available online for this figure.

controls (Figs. 3A and EV2D), which is consistent with human clinical cases (Wang, Gao et al, 2022).

To further investigate the neuropathology of RhoB^S73F/S73F rabbits, brain and spinal cord sections from 12-week-old rabbits were stained with H&E. Cell shrinkage, nuclear condensation, fragmentation, and granule cell loss were observed in the RhoB^S73F/S73F rabbits compared to WT controls (Fig. 3B), and the significant reductions in the myelin sheath of the external capsule, hippocampus, and geniculate nucleus in RhoB^S73F/S73F rabbits by using LFB staining (Fig. 3C). Moreover, the result of transmission electron microscopy (TEM) showed the destructive changes in the myelin sheath layer and subaxonal edema in RhoB^S73F/S73F rabbits (Fig. 3D).

Immunofluorescence analysis was performed to assess the extent of nerve cell damage in the cerebral cortex and hippocampus. Compared with WT controls, significant abnormalities in demyelination, neuronal reduction, astrocyte morphology, and microglial activation were determined in RhoB^S73F/S73F rabbits (Figs. 3E–G and EV2E). In addition, the significantly decreased GFAP around blood vessels (Fig. 3H) may suggest impairment of the blood-brain barrier in RhoB^S73F/S73F rabbits.

## Lipid homeostasis was disrupted in RhoB^S73F/S73F rabbits

In the TEM analysis, numerous lipid droplets accumulated around the damaged myelin sheath was observed in RhoB^S73F/S73F rabbits (Fig. 3D). Additionally, lipid droplet accumulation was also noted in neurons and astrocytes of RhoB^S73F/S73F rabbits, along with significant nuclear membrane depression, nuclear crinkling, and mitochondrial swelling (Fig. 4A). Furthermore, a significant increase in total cholesterol, cholesterol esters (Fig. 4B), triglyceride (TG) (Fig. 4C), and the lipid peroxidation product (malondialdehyde, MDA) (Fig. 4D) were observed in RhoB^S73F/S73F rabbit brain tissue, suggesting a potential imbalance in lipid metabolism.

Moreover, western blotting analysis revealed more cleaved GFAP protein in the brain tissue of RhoB^S73F/S73F rabbit compared to the WT control (Fig. 4E). The calpain1 protein, which could lead to the fragmentation of GFAP protein (Yang, Arja et al, 2022) and the degradation of demyelination (Shields, Schaecher et al, 1999), was significantly activated in RhoB^S73F/S73F rabbit brain tissue (Fig. 4F), which may suggest a calcium overload in the RhoB^S73F/S73F rabbit brain.

Previous studies showed that excessive calcium could lead to dysfunction in the endoplasmic reticulum (ER) (Lou, Wang et al,

2021), which plays a crucial role in the progression of various brain injuries (Hood, Zhao et al, 2018; Wang, Liu et al, 2022). In this study, the levels of GRP78 (a marker of ER stress), CHOP (a marker of ER stress-induced apoptosis), and cleaved Caspase3 (a marker of apoptosis) were significantly increased in the RhoB^S73F/S73F rabbit brain (Figs. EV3A,B and 4G). These findings may suggest that an imbalance in lipid homeostasis could lead to ER stress and apoptosis, potentially contributing to brain injury in RhoB^S73F/S73F rabbits.

## Activation of ACAT1 by the RhoB p.S73F mutation through LYN

To investigate the mutational pathogenesis, PyMol structural simulations of the RhoB protein were conducted, and the results revealed that the amino acid change at position 73 of the RhoB gene could impact its local surface potential (Figs. 5A and EV3C,D), thus consequently influencing its interaction with the target proteins. To confirm the impact of the mutation on the binding and function of the RhoB protein, an in vitro pull-down assay was conducted using purified recombinant proteins of RhoB WT and RhoB^S73F/S73F mutant (Schuster, Treitschke et al, 2012) (Fig. EV3E). The results demonstrated that the stronger binding affinity of 18 target proteins was observed in RhoB^S73F/S73F mutant than WT control (Fig. 5B), of which 13 proteins were regulated by tyrosinase phosphorylation, aligning with previous research indicating the role of RhoB in facilitating the activation of the tyrosine kinase SRC (Sandilands, Cans et al, 2004). Of note, the most significant differentially bound protein of ACAT1 (acetyl-CoA acetyltransferase 1), a tyrosine kinase substrate (Fig. 5C), may play crucial roles in the pathogenesis of RhoB p.S73F CP. Additionally, we also observed that RhoB binds to LYN (LYN proto-oncogene), a member of the SRC-family kinases (Table EV1).

Further studies showed significantly increased phosphorylation at the ACAT1 Y407 and LYN Y397 functional activation sites in RhoB^S73F/S73F rabbit brain tissue (Fig. 5D,E). To further validate the intrinsic link between ACAT1 and LYN, in vitro experiments were performed using the N2a cell line (WT) and the RhoB^S73F/S73F N2a cell line (RhoB^S73F/S73F). The result showed significantly enhanced ACAT1 phosphorylation in the N2a cells with overexpression of LYN and its activated mutant LYN Y397D (Fig. 5F). Additionally, the significant inhibition of ACAT1 phosphorylation was also determined in LYN-knockdown cells, which compared with control

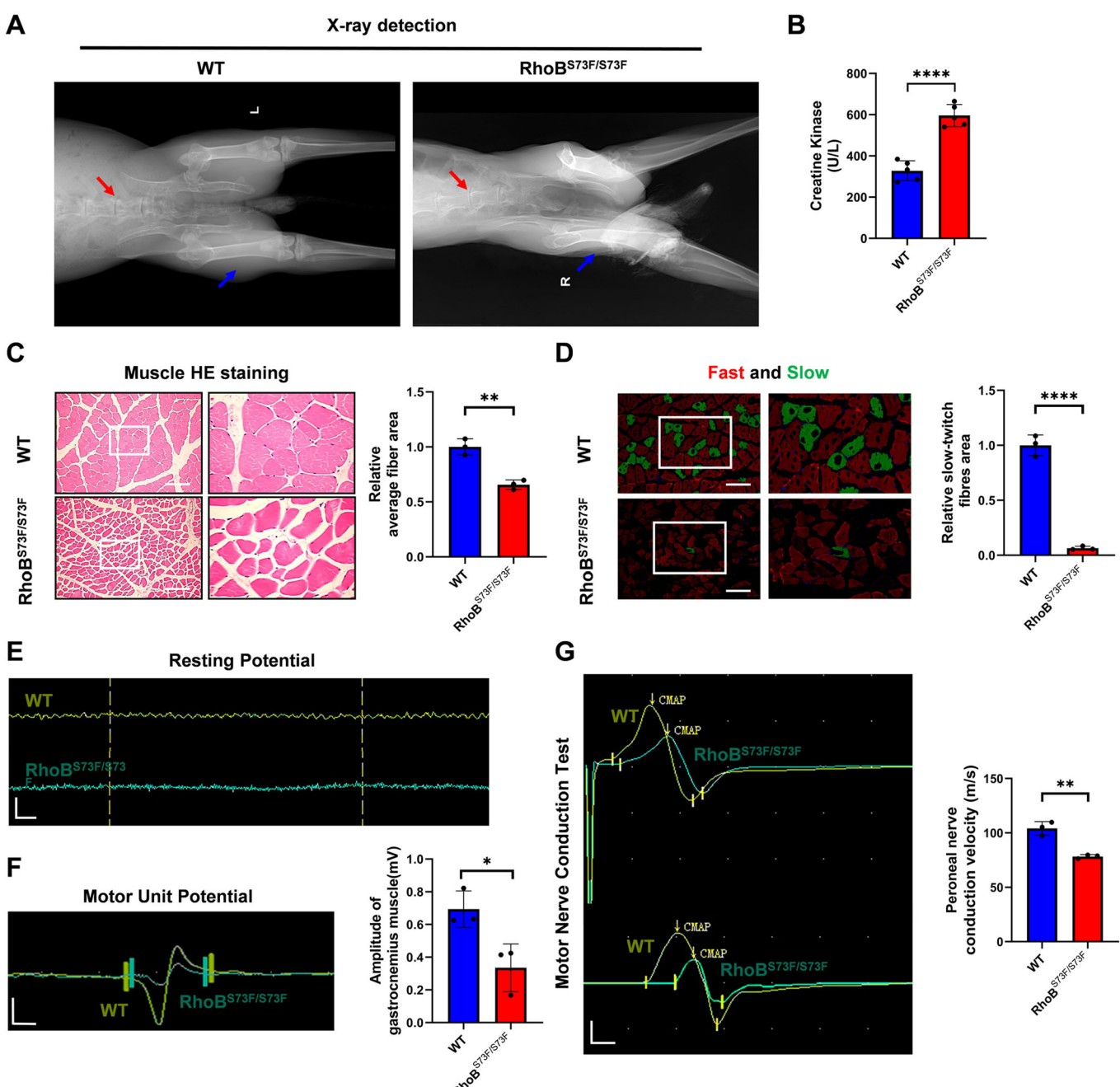

cells (Fig. 5G). We suggested that RhoB may regulate ACAT1 activation through LYN in this study.

## Disruption of lipid homeostasis by the RhoB p.S73F mutation could be alleviated by atorvastatin treatment

In this study, the activation of ACAT1 and dysregulation of lipid homeostasis were observed in RhoB$^{S73F/S73F}$ rabbit brain (Figs. 5D, 4A), which is consistent with previous studies that ACAT1 is an important acetyltransferase involved in lipid metabolism (Zhu, Gu et al, 2020). Thus, we hypothesized activation of ACAT1 could lead to dysregulation of lipid homeostasis in RhoB$^{S73F/S73F}$ rabbits.

To confirm this hypothesis, the stable cell lines of ACAT1-activated mutant overexpressed (ACAT1-M), the RhoB$^{S73F/S73F}$ mutated, and RhoB$^{S73F/S73F}$ treated with atorvastatin, a lipid-lowering agent to reduce cholesterol and triglyceride levels (RhoB$^{S73F/S73F}$ + ATV), were generated and utilized for subsequent experiments.

The results showed significantly increased ectopic free cholesterol, decreased free cholesterol in the cell membrane by Filipin staining (Fig. 6A), and increased lipid droplet by oil red O staining (Fig. EV3F) in the RhoB$^{S73F/S73F}$ cells and ACAT1-M cells compared with WT cells. While the significantly recovered free cholesterol in the cell membrane of the RhoB$^{S73F/S73F}$ + AVT cells compared with

◄ **Figure 2. The hind limb neurogenic lesion was observed in RhoB^S73F/S73F rabbits.**

(A) Representative X-ray detection images of 12-week-old WT controls (n = 3) and RhoB^S73F/S73F rabbits (n = 3). Radiographs reveal asymmetry in the L6 and L7 vertebral spaces, with a shift of the spine to the right, and bilateral lateral femoral muscle atrophy was observed in RhoB^S73F/S73F rabbits. Red arrows indicate rabbit lumbar vertebrae; blue arrows indicate rabbit lateral femoral muscle. (B) Serum detection of CK in 12-week-old WT controls (n = 5) and RhoB^S73F/S73F rabbits (n = 5). The significantly increased serum CK levels were observed in RhoB^S73F/S73F rabbits. See details on P values in Appendix Table S6. (C) HE staining images and relative mean fiber area analysis of left hind limb gastrocnemius muscle sections in 12-week-old WT controls (n = 3) and RhoB^S73F/S73F rabbits (n = 3). The significantly reduced mean fiber area was observed in RhoB^S73F/S73F rabbits. Scale bars: 200 μm. See details on P values in Appendix Table S7. (D) Immunofluorescence staining images and relative fiber area analysis for fast-twitch (Red) and slow-twitch (Green) of left hind limb gastrocnemius muscle sections in 12-week-old WT controls (n = 3) and RhoB^S73F/S73F rabbits (n = 3). The significant fiber type grouping was observed in RhoB^S73F/S73F rabbits. Nuclei DAPI-stained (Blue). Scale bars: 200 μm. See details on P values in Appendix Table S8. (E) Representative needle electromyography images of resting potential in the left hind limb gastrocnemius muscle from 12-week-old WT controls (n = 3) and RhoB^S73F/S73F rabbits (n = 3). Abnormal high-frequency discharges at the resting potential of the gastrocnemius muscle were observed in RhoB^S73F/S73F rabbits. Myoelectric Filters: 10 Hz to 5 kHz; Scanning rate and sensitivity: 20 ms/div, 100 μV/div. Scale bars: 10 ms; 50 μV. (F) Representative needle electromyography images and amplitude analysis of MUP in the left hind limb gastrocnemius muscle from 12-week-old WT controls (n = 3) and RhoB^S73F/S73F rabbits (n = 3). The significantly reduced MUP wave amplitude was observed in RhoB^S73F/S73F rabbits. Myoelectric Filters: 10 Hz to 5 kHz; Scanning rate and sensitivity: 10 ms/div, 500 μV/div. Scale bars: 2 ms; 200 μV. See details on P values in Appendix Table S9. (G) Representative CAMP images and peroneal nerve conduction velocity analysis in the lower limbs from 12-week-old WT controls (n = 3) and RhoB^S73F/S73F rabbits (n = 3). The significantly longer latencies, reduced wave amplitudes, and a significant slowing of nerve conduction velocity in the right hind limb were observed in RhoB^S73F/S73F rabbits. Myoelectric Filters: 20 Hz to 5 kHz; Stimulation current: 30 mA; Scanning rate and sensitivity: 2 ms/div, 5 mV/div. Scale bars: 1 ms; 2.5 mV. See details on P values in Appendix Table S10. Data information: Data represent different numbers (n) of biological replicates. In (B–D, F, G), data were presented as mean ± SD (Unpaired two-tailed Student's t-tests). *$P \leq 0.05$, **$P \leq 0.01$, ****$P \leq 0.0001$. Source data are available online for this figure.

the RhoB^S73F/S73F cells (Fig. 6A). In addition, the significantly increased TG (Fig. 6B), ROS (Fig. 6C), and MDA (Fig. 6D) levels in the RhoB^S73F/S73F cells than those in WT cells, consistent with the results in brain tissue of RhoB^S73F/S73F rabbits(Fig. 4C,D), while the significantly decreased those indicators in the RhoB^S73F/S73F + AVT cells compared with the RhoB^S73F/S73F cells (Fig. 6B–D), which may contribute to the stability of cell membrane permeability (Yang, Wang et al, 2019).

Furthermore, significantly increased intracellular calcium levels (Fig. 6E), calpain1 activation (Fig. 6F), GRP78 and CHOP protein levels (Fig. 6G,H), and decreased cell survival (Fig. 6I) in the RhoB^S73F/S73F cells than WT cells, while the significantly restored those indicators in the RhoB^S73F/S73F + AVT cells compared with the RhoB^S73F/S73F cells (Fig. 6E–I).

In summary, the RhoB p.S73F mutation may enhance the interaction with LYN and potentially result in continuous activation of the ACAT1 protein. Consequently, this activation could lead to lipid accumulation and disruption of lipid homeostasis, which might induce the generation of ROS in mitochondria (Zhang, Zhang et al, 2022). Moreover, lipid peroxidation could alter membrane fluidity and permeability, ultimately resulting in calcium overload and subsequent damage to neuronal cells and myelin (Fig. 6J).

## Discussion

Although the RhoB p.S73F mutation has been identified as a candidate gene for CP (Jin et al, 2020), the mechanism by which the RhoB p.S73F mutation causes CP is currently unclear. In this study, we provide the first animal model evidence to mimic the RhoB p.S73F mutation, which showed the typical phenotype of neuromotor impairment, hind limb neurogenic lesions, brain tissue damage, mechanism of the dysregulated lipid homeostasis and neurological damage via LYN-ACAT1 in this rabbit model. Therefore, this novel rabbit model could be used as a model for recapitulating CP, and provide further insights between RhoB and lipid metabolism, and therapeutic targets for human CP.

In newborns at high risk for CP, MRI scans typically reveal damage to white matter, structural damage to deep gray matter, and developmental brain malformations (Novak et al, 2017; Spittle,

Morgan et al, 2018). In RhoB^S73F/S73F rabbits, pathological damage was primarily observed in the white matter of the brain, which is consistent with reports of human patients with RhoB-mutant CP (Jin et al, 2020). We speculate that this may be because of the higher expression level of RhoB in white matter than in gray matter. Furthermore, the onset of RhoB^S73F/S73F rabbits' paralysis symptoms aligned with important developmental stages in mammalian white matter (Huang, Bhaduri et al, 2020), both during childhood and later stages, suggesting an important temporal node for early intervention in CP.

Whereas CP results from a primary injury in the central nervous system (CNS), clinical symptoms are observed in the peripheral neuromuscular system, skeletal muscles in particular (Graham, Rosenbaum et al, 2016). Noteworthy, peripheral neuropathy is a rare phenotype in individuals with CP (Lim, Smith et al, 2014). Our studies demonstrated signs of peripheral neuropathy, specifically damage to the common peroneal nerve in RhoB^S73F/S73F rabbits. Further investigation of this occurrence in human patients with RhoB-mutant CP could provide valuable insights into the disease prognosis.

Rho GTPases are a crucial link between upstream regulatory proteins and downstream effector proteins, and their spatiotemporal functions rely heavily on conformational changes in their molecular switches (Vetter and Wittinghofer, 2001). The RhoB p.S73F mutation is located in the Switch II region, and the serine at position 73 (S73) is hypothesized to be a phosphorylation site, based on studies of GTPases homologous family (Kwon, Kwon et al, 2000; Lehman, Van Laere et al, 2012). In this study, structural simulation results indicated that the RhoB p.S73F mutation alters the polarity and hydrogen bonding of the amino acid at that position (Figs. 5A and EV3C,D), affecting the conformation of the molecular switch domain, and disrupting the ability of RhoB to effectively regulate downstream signals. Additionally, the stronger binding affinity of the RhoB p.S73F protein to ARHGEF2 as well as RASA4 and RANBP1 was observed, according to the pull-down and mass spectrometry (MS) analysis (Data availability). This finding is consistent with Jin's (2020) biochemical investigations, which demonstrated that the p.S73F mutation significantly increased the sensitivity of the RhoB protein to guanine nucleotide exchange factors (GAPs) and GTPase activating proteins (GEFs), as

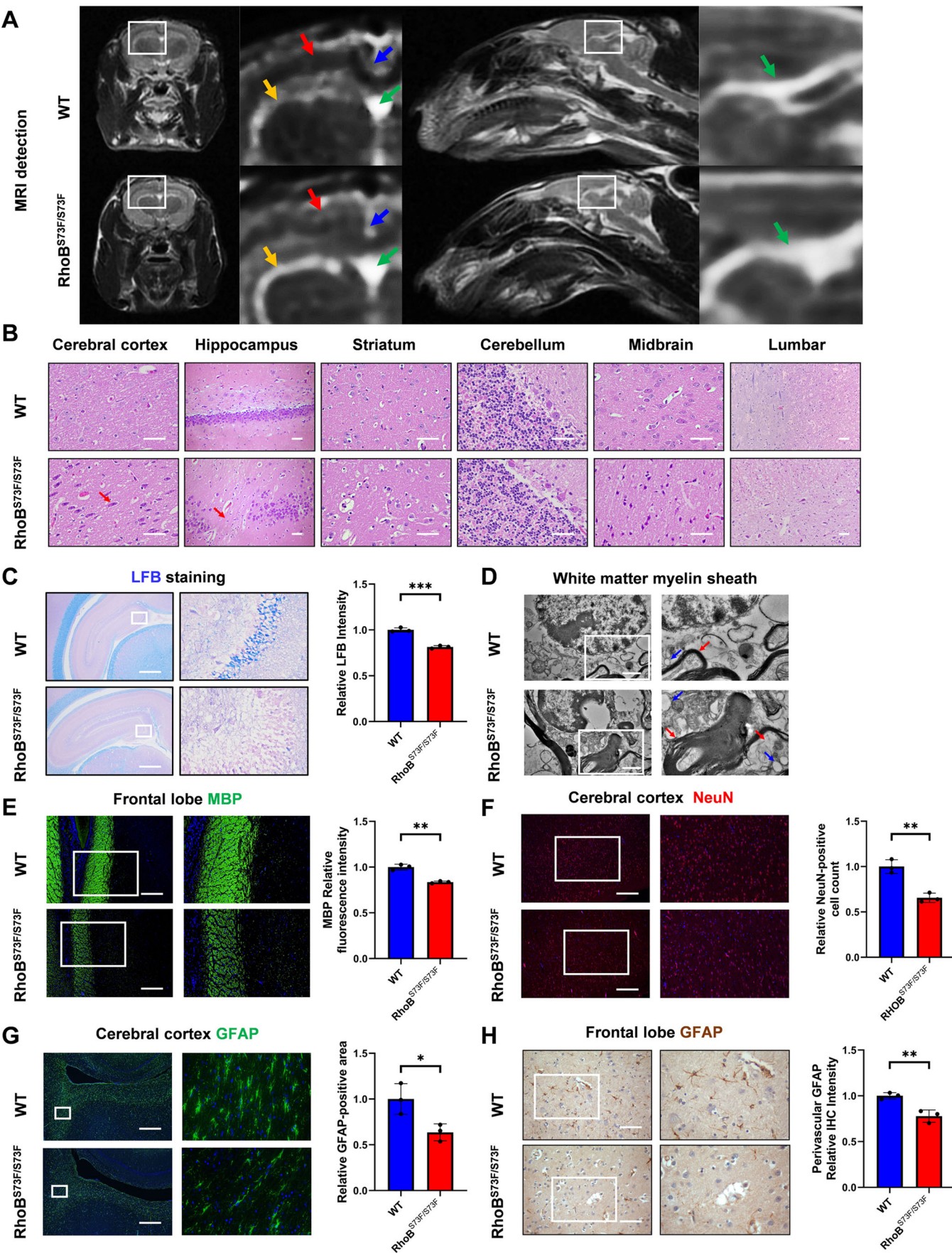

**Figure 3. Brain tissue damage was observed in RhoB$^{S73F/S73F}$ rabbits.**

(A) MRI detection images of 12-week-old WT controls ($n = 3$) and RhoB$^{S73F/S73F}$ rabbits ($n = 3$). Several abnormalities were observed in RhoB$^{S73F/S73F}$ rabbits. Red arrows indicate the white matter of the rabbit brain; yellow arrows indicate the lateral ventricles of the rabbit brain; blue arrows indicate the corpus callosum of the rabbit brain; green arrows indicate the third ventricle of the rabbit brain. (B) HE staining images of brain sections in 12-week-old WT controls ($n = 3$) and RhoB$^{S73F/S73F}$ rabbits ($n = 3$). Significant signs of neuronal loss, cell shrinkage, nuclear condensation, and fragmentation, particularly in the cerebral cortex and hippocampus, were observed in RhoB$^{S73F/S73F}$ rabbits. Red arrows indicate damaged and lost neurons. Scale bars: 50 µm. (C) LFB staining images and relative intensity (OD value) analysis of brain sections in 12-week-old WT controls ($n = 3$) and RhoB$^{S73F/S73F}$ rabbits ($n = 3$). Significant demyelination was observed in RhoB$^{S73F/S73F}$ rabbits. Scale bars: 1000 µm. See details on $P$ values in Appendix Table S11. (D) TEM images of the brain in 12-week-old WT controls ($n = 3$) and RhoB$^{S73F/S73F}$ rabbits ($n = 3$). The presence of destructive changes in the myelin sheath layer and myelinated nerve fibers, along with subaxonal edema, were observed in RhoB$^{S73F/S73F}$ rabbits. Red arrows indicate myelin; blue arrows indicate lipid droplets. Scale bars: 1 µm. (E) Immunofluorescence staining images and relative fluorescence intensity analysis of MBP (Green) in brain sections from 12-week-old WT controls ($n = 3$) and RhoB$^{S73F/S73F}$ rabbits ($n = 3$). The significantly reduced MBP protein was observed in RhoB$^{S73F/S73F}$ rabbits. Nuclei DAPI-stained (Blue). Scale bars: 200 µm. See details on $P$ values in Appendix Table S12. (F) Immunofluorescence staining images and relative fluorescence intensity analysis of NeuN (Red) in brain sections from 12-week-old WT controls ($n = 3$) and RhoB$^{S73F/S73F}$ rabbits ($n = 3$). The significantly reduced neurons were observed in RhoB$^{S73F/S73F}$ rabbits. Nuclei DAPI-stained (Blue). Scale bars: 500 µm. See details on $P$ values in Appendix Table S13. (G) Immunofluorescence staining images and relative positive area analysis of GFAP (Green) in brain sections from 12-week-old WT controls ($n = 3$) and RhoB$^{S73F/S73F}$ rabbits ($n = 3$). The significantly damaged astrocytes were observed in RhoB$^{S73F/S73F}$ rabbits compared to the WT controls. Nuclei DAPI-stained (Blue). Scale bars: 500 µm. See details on $P$ values in Appendix Table S14. (H) Immunohistochemical staining images and perivascular relative IHC intensity (OD value) analysis of GFAP in brain sections from 12-week-old WT controls ($n = 3$) and RhoB$^{S73F/S73F}$ rabbits ($n = 3$). A significant decrease in perivascular GFAP protein was observed in RhoB$^{S73F/S73F}$ rabbits. Scale bars: 100 µm. See details on $P$ values in Appendix Table S15. Data information: Data represent different numbers ($n$) of biological replicates. In (C, E–G, H), data were presented as mean ± SD (Unpaired two-tailed Student's $t$-tests). *$P \le 0.05$, **$P \le 0.01$, ***$P \le 0.001$. Source data are available online for this figure.

well as its binding ability to the active form of the Rho effector Rhotekin (Jin et al, 2020). Additionally, the RhoB protein showed a significant increase in the RhoB$^{S73F/S73F}$ rabbit brain (Fig. EV3G), which might be associated with increased intracellular ROS and calcium ion levels as reported in previous studies (Kajimoto, Hashimoto et al, 2007; Sakurada, 2003).

Our study showed the aggregation of lipid droplets observed in RhoB$^{S73F/S73F}$ rabbits' neuronal cells. However, no published data were found regarding the association between CP and lipids currently. Previous studies showed that patients with CP have higher mortality of cardiovascular disease-related than does the general population, attributing it to the lack of physical activity and sedentary lifestyles of CP patients (Mcphee, Claridge et al, 2018; Ryan, Crowley et al, 2014). Nevertheless, our results implied that abnormalities in lipid metabolism could play an important role in the initiation of CP brain injury, offering potential opportunities for early intervention strategies.

In summary, the first mammalian genetic CP model that precisely mimics a human pathological point mutation in the RhoB gene was successfully constructed in this study. The molecular pathological mechanism based on this model revealed that RhoB p.S73F mutation could disrupt the regulation of lipid homeostasis by activating the LYN-ACAT1 pathway, leading to neuronal and white matter damage, providing valuable insights for potential therapeutic strategies in human CP.

## Methods

### Reagents and tools table

| Reagent/Resource | Reference or source | Identifier or Catalog Number |
|---|---|---|
| **Experimental models** | | |
| New Zealand white rabbits | Changsheng | SCXK20200001 |
| Neuro-2a cell | ATCC | #CCL-131 |
| SH-SY5Y cell | ATCC | #CRL-2266 |

| Reagent/Resource | Reference or source | Identifier or Catalog Number |
|---|---|---|
| **Recombinant DNA** | | |
| pUC57-T7-sgRNA | Addgene | #51306 |
| SpG-BE4max | Addgene | #139998 |
| pGEX-6P-1 | Amersham | 27-4597-01 |
| pGEX-6P-1-RhoB | This study | |
| pGEX-6P-1-RhoB$^{S73F/S73F}$ | This study | |
| pGL3-U6-sgRNA | Addgene | #51133 |
| mEGFP-N1 | Addgene | #54767 |
| mEGFP-N1-LYN | This study | |
| mEGFP-N1-ACAT1 | This study | |
| pmCherry-C1 | Takara | 632524 |
| pmCherry-C1-ACAT1 | This study | |
| mEGFP-N1-LYN Y397D | This study | |
| mEGFP-N1-ACAT1-M | This study | |
| pmCherry-C1-ACAT1-M | This study | |
| **Antibodies** | | |
| mAb=monoclonal; pAb=polyclonal | | |
| Rabbit pAb anti-myosin-1, IF (1: 400) | Servicebio | GB112130 |
| Rabbit pAb anti-myosin-7, IF (1: 400) | Servicebio | GB111857 |
| Mouse mAb anti-GAPDH, WB (1: 2000) | Proteintech | 60004-1-Ig |
| Rabbit pAb anti-GAPDH, WB (1: 2000) | Proteintech | 10494-1-AP |
| Mouse mAb anti-beta tubulin, WB (1: 2000) | Biozellen | B-IO-10032 |
| Rabbit pAb anti-caspase3/p17/p19, WB (1: 2000) | Proteintech | 19677-1-AP |
| Rabbit pAb anti-GFAP polyclonal antibody, WB (1: 2000), IF (1: 500), IHC (1: 500) | Proteintech | 16825-1-AP |
| Rabbit pAb anti-Iba1, IF (1: 500) | Abcom | ab108539 |
| Rabbit pAb anti-MBP, IF (1: 300) | Proteintech | 10458-1-AP |

| Reagent/Resource | Reference or source | Identifier or Catalog Number |
|---|---|---|
| Mouse recombinant pAb anti-NeuN, IF (1: 300) | Abcom | ab279296 |
| Rabbit pAb anti-RhoB, IF (1: 400) | Proteintech | 14326-1-AP |
| Rabbit pAb anti-ACAT1, WB (1: 2000) | Proteintech | 16215-1-AP |
| Rabbit pAb anti-ACAT1(P-Tyr407), WB (1: 1000) | Sabbiotech | #SAB491P |
| Mouse mAb anti-LYN, WB (1: 1000) | Proteintech | 60211-1-Ig |
| Rabbit pAb anti-Lyn (P-Tyr397), WB (1: 2000) | Bioss | bs-3257R |
| Mouse mAb anti-phosphotyrosine (P-Y99), WB (1: 500) | Santa | sc-7020 |
| Rabbit pAb anti-calpain1, WB (1: 1000) | Proteintech | 10538-1-AP |
| Rabbit pAb anti-GRP78/BIP, WB (1: 2000) | Proteintech | 11587-1-AP |
| Rabbit pAb anti-CHOP, IF (1: 500) | Proteintech | 15204-1-AP |
| Mouse mAb anti-CHOP, IF (1: 500) | Proteintech | 66741-1-Ig |
| Goat anti-mouse IgG HRP, WB (1: 10,000) | Proteintech | SA00001-1 |
| Goat anti-rabbit IgG HRP, WB (1: 10,000), IHC (1: 1000) | Proteintech | SA00001-2 |
| Goat anti-mouse CoraLite488, IF (1: 300) | Proteintech | SA00013-1 |
| Goat anti-rabbit CoraLite488, IF (1: 300) | Proteintech | SA00013-2 |
| Goat anti-mouse CoraLite594, IF (1: 300) | Proteintech | SA00013-3 |
| Goat anti-rabbit CoraLite594, IF (1: 300) | Proteintech | SA00013-4 |
| **Oligonucleotides and other sequence-based reagents** | | |
| PCR primers | This study | Methods and Protocols |
| **Chemicals, enzymes, and other reagents** | | |
| MAXIscript T7 Kit | Ambion | AM1314 |
| miRNeasy Mini Kit | Qiagen | 217004 |
| HiScribe T7 ARCA mRNA Kit | NEB | E2060S |
| RNeasy Mini Kit | Qiagen | 74106 |
| E. coli BL21 (DE3) | Tiangen | CB105 |
| GST-tag Protein Purification Kit | Beyotime | P2262 |
| RIPA lysis buffer | MeilunBio | MA0151 |
| Hematoxylin and eosin | Solarbio | G1120 |
| LFB staining solution | Servicebio | G1044 |
| SP Kit | Solarbio | SP0041 |
| pH 6.0 citrate buffer | Servicebio | G1202 |
| Antifade mounting medium containing DAPI I | Beyotime | P0131 |
| DMEM | Gibco | 11965092 |
| FBS | Clark Bioscience | FB25015 |
| Puromycin | MeilunBio | MA0318 |
| PVDF membranes | Millipore | IPVH00010 |
| Skim milk | Boster | AR0104 |
| ECL detection reagents | MeilunBio | MA0186 |

| Reagent/Resource | Reference or source | Identifier or Catalog Number |
|---|---|---|
| Micro TC Content Assay Kit | Solarbio | BC1985 |
| Micro FC Content Assay Kit | Solarbio | BC1895 |
| Modified Oil Red O Stain Kit | Solarbio | G1263 |
| Micro TG Assay Kit | Abbkine | KTB2200 |
| Micro (MDA Assay kit | Abbkine | KTB1050 |
| Filipin | MeilunBio | MB1848 |
| Reactive Oxygen Species Assay Kit | MeilunBio | MA0219 |
| Cell Counting Kit-8 | APExBIO | K1018 |
| Fura-2 AM | MeilunBio | MA0194 |
| **Software** | | |
| Pawfit | Latsen | |
| Olympus cellSens software | cellSens | |
| ImageJ | NIH | |
| GraphPad Prism 8 | Graphpad | |
| **Other** | | |
| Tensiometer | Sanliang | |
| Digital radiography system | Varian | |
| SIGNA Creator MRI scanner | GE Healthcare | |
| NeuroExam M-800 | Medcom | |
| Olympus IX73 | Olympus | |
| Tanon 5200 | Tanon | |
| Infinite 200 PRO | Tecan | |

## Methods and protocols

### Animals and ethics statement

New Zealand white rabbits were used in this study, which was obtained from Changsheng Biotechnology (China, SCXK20200001). The rabbits for experiments in this study were kept and studied in strict accordance with the Jilin University Laboratory Animal Welfare Ethical Review Guidelines and the ARRIVE guidelines. Rabbits were kept on a 12-h light and dark cycle, with lights turning on at 7 a.m. Food and water were provided ad libitum. The Institutional Animal Care and Use Committee of Jilin University, China, provided ethical approval for this research (SY202301001).

### Vector construction and in vitro transcription

The annealed sgRNA oligonucleotides were cloned into the BbsI site of the pUC57-T7-sgRNA cloning vector from Addgene (#51306). The sgRNA was then amplified and transcribed in vitro using the MAXIscript T7 Kit (Ambion, AM1314) and purified using the miRNeasy Mini Kit (Qiagen, 217004). The SpG-BE4max plasmid was obtained from Addgene (#139998), which was linearized with NotI and transcribed in vitro using the HiScribe T7 ARCA mRNA Kit (NEB, E2060S) and purified using the RNeasy Mini Kit (Qiagen, 74106).

### Embryo microinjection and embryo transfer

The procedures for embryo microinjection and transfer are according to the previously published protocol (Liu, Chen et al,

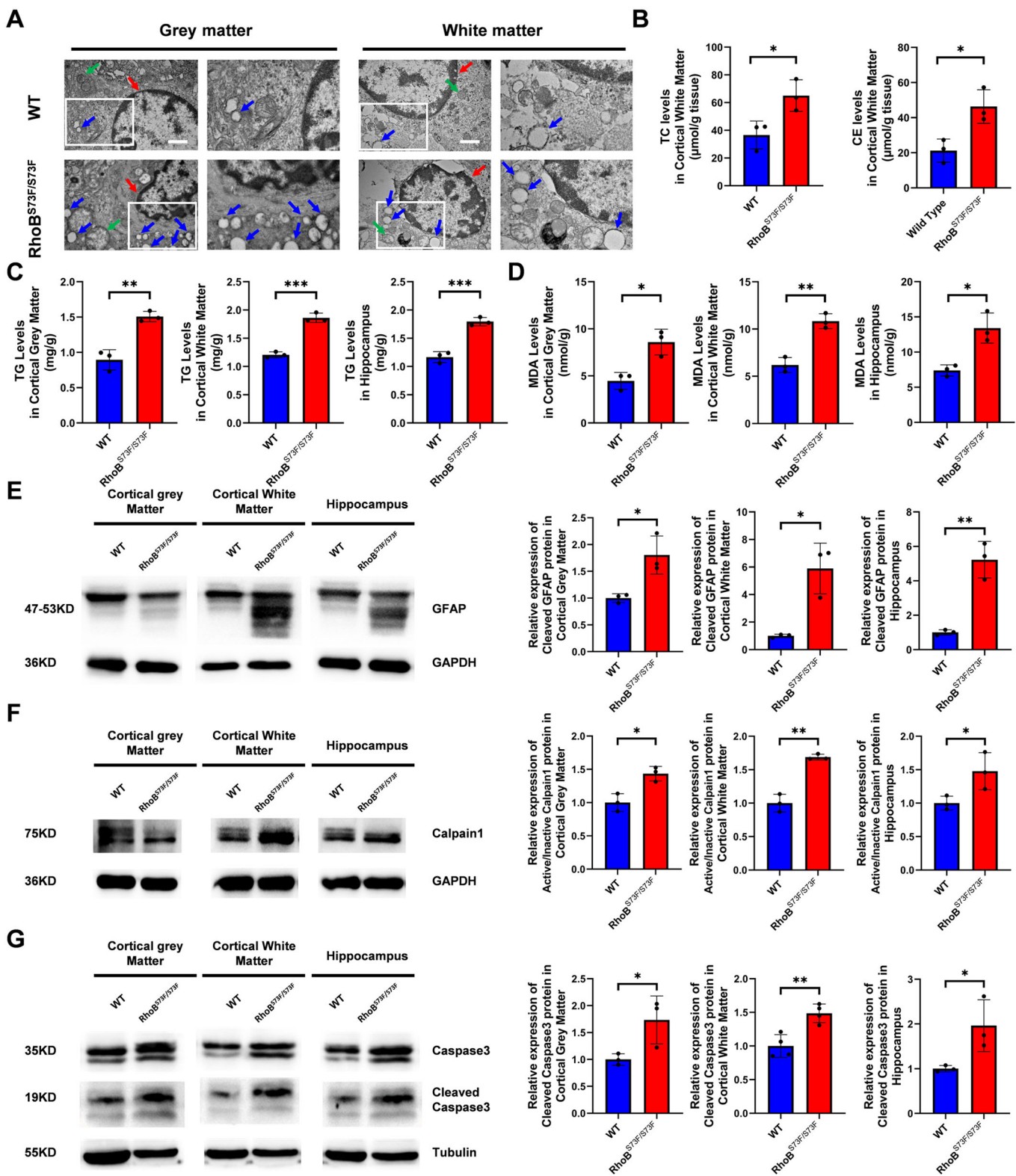

2018). Briefly, a mixture of Cas9 mRNA (200 ng/μL) and sgRNA (50 ng/μL) was microinjected into the cytoplasm of zygotes. Approximately 30–50 injected zygotes were then transferred into the oviducts of surrogate rabbits.

**Mutation detection in pups by PCR and sequencing**

The genomic DNA of newborn rabbits was extracted from ear clips and analyzed by PCR genotyping and Sanger sequencing

◄ **Figure 4. Lipid homeostasis was disrupted in RhoB$^{S73F/S73F}$ rabbits.**

(A) TEM images of the brain in 12-week-old WT controls ($n = 3$) and RhoB$^{S73F/S73F}$ rabbits ($n = 3$). Significant nuclear membrane depression, nuclear crinkling, mitochondrial swelling, and lipid droplet accumulation were observed in RhoB$^{S73F/S73F}$ rabbits compared to the WT controls. Red arrows indicate the nucleus; green arrows indicate the mitochondria; and blue arrows indicate the lipid droplets. Scale bars: 1 μm. (B) TC levels analysis and CE levels analysis of brain white matter in 12-week-old WT controls ($n = 3$) and RhoB$^{S73F/S73F}$ rabbits ($n = 3$). The significantly upregulated TC and CE were observed in RhoB$^{S73F/S73F}$ rabbit brain white matter compared to the WT control. See details on $P$ values in Appendix Table S16. (C) TG levels analysis of the brain in 12-week-old WT controls ($n = 3$) and RhoB$^{S73F/S73F}$ rabbits ($n = 3$). The significantly upregulated TG levels were observed in RhoB$^{S73F/S73F}$ rabbits. See details on $P$ values in Appendix Table S17. (D) MDA levels analysis of the brain in 12-week-old WT controls ($n = 3$) and RhoB$^{S73F/S73F}$ rabbits ($n = 3$). The significantly upregulated MDA levels were observed in RhoB$^{S73F/S73F}$ rabbits compared to the WT controls. See details on $P$ values in Appendix Table S18. (E) WB detection and relative expression analysis of cleaved GFAP in the brain from 12-week-old WT controls ($n = 3$) and RhoB$^{S73F/S73F}$ rabbits ($n = 3$). The significantly upregulated cleaved GFAP protein was observed in RhoB$^{S73F/S73F}$ rabbits. See details on $P$ values in Appendix Table S19. (F) WB detection and relative expression analysis of active Calpain1 in the brain from 12-week-old WT controls ($n = 3$) and RhoB$^{S73F/S73F}$ rabbits ($n = 3$). The significantly activated Calpain1 protein was observed in RhoB$^{S73F/S73F}$ rabbits. See details on $P$ values in Appendix Table S20. (G) WB detection and relative expression analysis of cleaved caspase3 in the brain from 12-week-old WT controls and RhoB$^{S73F/S73F}$ rabbits (Cortical gray and hippocampus groups $n = 3$, white matter group $n = 4$). The significantly upregulated cleaved caspase3 protein was observed in RhoB$^{S73F/S73F}$ rabbits. See details on $P$ values in Appendix Table S21. Data information: Data represent different numbers ($n$) of biological replicates. In (B–G), data were presented as mean ± SD (Unpaired two-tailed Student's $t$-tests). *$P \leq 0.05$, **$P \leq 0.01$, ***$P \leq 0.001$. Source data are available online for this figure.

using primers (Forward, 5′- GGTGAGCAGTGAGCGAAG -3′; Reverse, 5′- AGGTAGTCGTAGGCTTGGAT -3′).

### Body weight and survival curve
Survival of WT and RhoB$^{S73F/S73F}$ rabbits was recorded daily and body weight was measured every two weeks. All data are expressed as mean ± SD, and at least three rabbits from each group were used in all experiments.

### Gait analysis and open field test
Gait analysis was performed by immersing the rabbit's hind paws in blue dye, while the fore paws in red dye. The data were collected and analyzed for the distance of movement of the rabbit's hind feet, the horizontal distance, and the distance between the left and right limbs.

Spontaneous locomotor activity was assessed using the open field test with Pawfit (Latsen Technology), a smart GPS and activity pet tracker.

### Muscle tone test
Muscle tone was assessed by measuring the peak force generated during flexion of the rabbit's hind limb. An electric stimulus was applied safely to induce flexion movements in the rabbit hind limb, with the resulting forces recorded to quantify the level of flexion force in rabbits by tensiometer (Sanliang, SF-20N). Frequency: 3.1 Hz; Pulse width: 200 μs; Current: 30 mA.

### Buried food pellet test
The buried food pellet test was carried out as described previously (Nathan, Yost et al, 2004). Rabbits were food deprived for 12 h before the test. The latency to find food pellet was monitored. If the rabbits failed to find the buried food within 60 min, the test was stopped, and the latency score was recorded as 60 min.

### Skeletal radiography and MRI
Skeletal radiography was performed using a digital radiography system (Varian) at the Jilin University Veterinary Hospital. Supine position: 50 KV, 10 mAs.

MRI scans were performed in the Campo Medical Imaging Center using a SIGNA Creator MRI scanner (GE Healthcare). MRI scans were performed in a standardized manner and interpreted with reference to studies by Flodmark (Bax, Tydeman et al, 2006); Xu (Jin, Pan et al, 2018), and Gratacos (Muñoz-Moreno et al, 2013)

et al, Coronal T2 FSE: TR 3000 ms, TE 100 ms, Thk 4 mm; Sagittal T2 FSE: TR 2300 ms, TE 100 ms, Thk 4 mm.

### Pull down and mass spectrometry
To express the GST-tag, GST-RhoB, and GST-RhoB$^{S73F/S73F}$ proteins, the target fragments were integrated into the pGEX-6P-1 vector, and the plasmids were transformed into *E. coli* BL21 (DE3) bacteria (Tiangen, CB105). The bacteria were then incubated at 16 °C for 16 h with the addition of 0.2 mM IPTG to induce protein expression. The GST-tag and GST-RhoB recombinant proteins were purified using a GST-tag Protein Purification Kit (Beyotime, P2262) according to the manufacturer's instructions. SH-SY5Y cells were washed with cold PBS and lysed with RIPA lysis buffer (MeilunBio, MA0151). The cell lysate supernatant was then incubated with glutathione agarose beads bound to purified protein for 3 h at 4 °C. Finally, after washing and elution, the total protein, including decoy and target proteins, was obtained for subsequent MS and interaction analysis. MS and protein identification services were provided by the PTM Biolabs Inc. in Table EV1.

### Electromyography
All muscle electrophysiological measurements were recorded and analyzed using an electromyography/evoked potential instrument, NeuroExam M-800 (Medcom). Needle electromyography and compound muscle action potentials assays were performed on the gastrocnemius and common peroneal nerve of WT and RhoB$^{S73F/S73F}$ rabbits according to the manufacturer's instructions.

### Histology
Nerve tissue, including brain and spinal cord, was collected from WT and RhoB$^{S73F/S73F}$ rabbits. The tissue was fixed, dehydrated, embedded, and 5 μm paraffin sections were cut from corresponding sections of each nerve region selected for subsequent histological examination.

For hematoxylin and eosin (HE) stain, paraffin sections were stained with hematoxylin and eosin (Solarbio, G1120) and examined by microscopy; for luxol fast blue (LFB) myelin stain, paraffin sections were stained with LFB staining solution (Servicebio, G1044) and examined by microscopy; for immunohistochemical (IHC) stain, paraffin sections were stained with SP Kit (Solarbio, SP0041) using appropriate primary antibodies and examined by microscopy; for immunofluorescence (IF) stain,

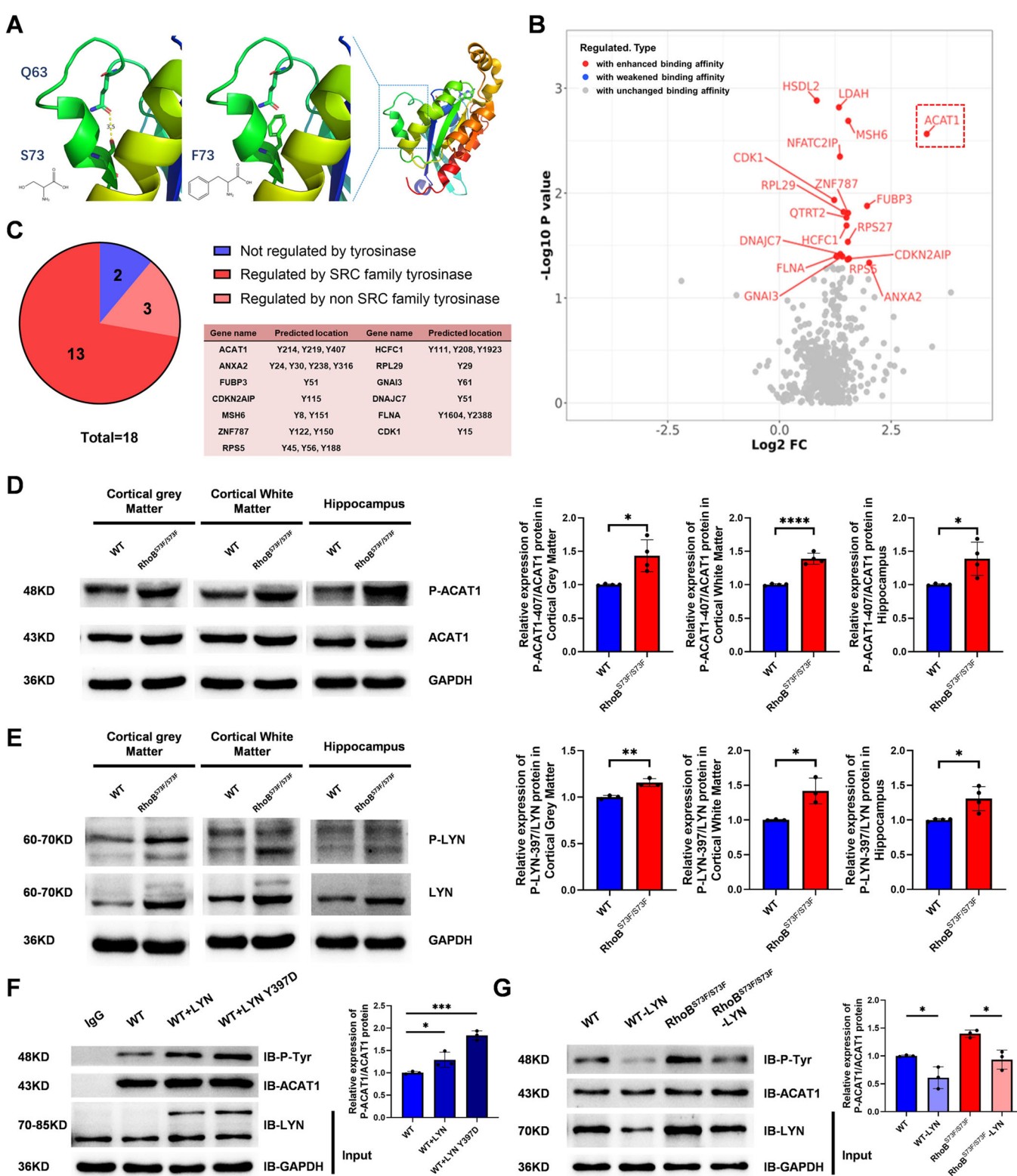

paraffin sections were deparaffinized and rehydrated, and antigen retrieval was performed using pH 6.0 citrate buffer (Servicebio, G1202). Tissue sections are incubated in 10% goat serum blocking solution, then incubated overnight at 4 °C with primary antibodies,

followed by 1 h of incubation at room temperature with appropriate fluorescently labeled secondary antibodies. Sections were then rinsed, mounted, and cover slipped with an antifade mounting medium containing DAPI (Beyotime, P0131).

◄ **Figure 5. Activation of ACAT1 by the RhoB p.S73F mutation through LYN.**

(A) Schematic comparison of the molecular structures in WT control and p.S73F mutant group of human-derived RhoB protein. The loss of internal hydrogen bonds was observed in the RhoB$^{S73F/S73F}$ protein compared to the WT control. (B) Volcano map of the mass spectra differentially bound proteins for pulled down by RhoB WT ($n = 3$) and RhoB$^{S73F/S73F}$ mutant proteins ($n = 3$). A stronger binding affinity to target proteins was observed in the RhoB$^{S73F/S73F}$ group (Unpaired two-tailed Student's $t$-tests). Red dots represent 18 bound proteins with enhanced binding affinity (FC ≥1.7 and $P < 0.05$); blue dots represent 0 bound proteins with weakened binding affinity; FC fold-change. (C) Phosphorylation analysis of differentially bound proteins. 13 out of 18 differentially binding proteins were observed to be regulated by SRC-family tyrosinase. Data were obtained from the PhosphoSitePlus (https://www.phosphosite.org/homeAction). Phosphomotif analysis were obtained from the Human Protein Reference Database (http://hprd.org/PhosphoMotif_finder). (D) WB detection and relative expression analysis of P-ACAT1 in the brain from 12-week-old WT controls ($n = 4$) and RhoB$^{S73F/S73F}$ rabbits ($n = 4$). The significantly activated ACAT1 protein was observed in RhoB$^{S73F/S73F}$ rabbits. See details on $P$ values in Appendix Table S22. (E) WB detection and relative expression analysis of P-LYN in the brain from 12-week-old WT controls and RhoB$^{S73F/S73F}$ rabbits (Cortical gray and white matter groups $n = 3$, hippocampus group $n = 4$). The significantly activated LYN protein was observed in RhoB$^{S73F/S73F}$ rabbits. See details on $P$ values in Appendix Table S23. (F) Co-IP-WB detection and relative expression analysis of P-ACAT1 in LYN overexpressing cells ($n = 3$ per group). A significant enhancement of ACAT1 phosphorylation was observed in the LYN and its activated mutant LYN Y397D overexpressing cells. See details on $P$ values in Appendix Table S24. (G) Co-IP-WB detection and relative expression analysis of P-ACAT1 in LYN-knockdown cells ($n = 3$ per group). A significant inhibition of ACAT1 phosphorylation was observed in the LYN-knockdown cells. See details on $P$ values in Appendix Table S25. Data information: Data represent different numbers ($n$) of biological replicates. In (D, E) data were presented as mean ± SD (Unpaired two-tailed Student's $t$-tests). In (F, G), data were presented as mean ± SD (One-way ANOVA). *$P ≤ 0.05$, **$P ≤ 0.01$, ***$P ≤ 0.001$, ****$P ≤ 0.0001$. Source data are available online for this figure.

Sections were imaged using an Olympus IX73 inverted microscope (Olympus, Tokyo, Japan) and Olympus cellSens software (cellSens Standard 1.9). Three randomly selected fields of view imaged at 200× magnification within the region of interest of each brain section on anatomically defined coronal sections were quantitatively analysed by ImageJ software (ImageJ 1.5, NIH, USA) by an observer blind. The antibodies used in this study are listed in the Reagents and tools table.

### Cell culture and selection of stable cell line

The mouse neuroblastoma Neuro-2a (N2a) cell line was procured from the American Type Culture Collection (USA, #CCL-131). Cell lines were authenticated with STR profiling and tested negative for mycoplasma contamination prior to the study. N2a and stable RhoB$^{S73F/S73F}$ cells were cultured in DMEM (Gibco, 11965092) containing 10% fetal bovine serum (FBS, Clark Bioscience).

To construct the RhoB$^{S73F/S73F}$ mutant N2a cell line, sgRNA targeting the murine-derived locus was inserted into the backbone vector from Addgene (#51133). Next, it was cotransfected into N2a cells with the SpG-BE4max plasmid. 48 h later, the stable cell line was selected with 1 μg/ml puromycin (MeilunBio, MA0318) for 1 week. Fluorescence microscopy was used to confirm the transfection efficiency. Monoclonal cells were screened by the limited dilution method and genotyped by Sanger sequencing.

### Overexpression and knockout plasmid construction

The potential phosphorylation sites of LYN Y397, ACAT1 Y214, Y219, and Y407 have been described previously (Fan, Lin et al, 2016, Lupo, Tibaldi et al, 2016).

The LYN-EGFP overexpression plasmid (mEGFP-N1-LYN) and ACAT1-EGFP overexpression plasmid (mEGFP-N1-ACAT1) were constructed by cloning the corresponding coding sequences into the mEGFP-N1 vector from Addgene (#54767); the ACAT1-mCherry overexpression plasmid (pmCherry-C1-ACAT1) were constructed by cloning the corresponding coding sequences into the pmCherry-C1 vector; the LYN Y397D-EGFP overexpression plasmid (mEGFP-N1-LYN Y397D), ACAT1 Y214D, Y219D, Y407D-EGFP overexpression plasmid (mEGFP-N1-ACAT1-M), and ACAT1 Y214D, Y219D, Y407D-EGFP overexpression plasmid (pmCherry-C1-ACAT1-M) were constructed using the Fast Site-Directed Mutagenesis Kit (Tiangen, KM101).

For gene knockout, Cas-Designer (http://www.rgenome.net/cas-designer/) was used to design the sgRNAs (sgRNA1: TCTATTC-CAACAGGAAATAT; sgRNA2: CCAGTAAGTAGACTAGTCTC; sgRNA3: GTGGCCTTATACCCTTATGA; sgRNA4: GCTGGG ACCTACTTGCTGCT); sgRNAs were inserted into the backbone vector from Addgene (#51133).

### Western blot

Western blot analysis was performed as previously described (Liu, Xu et al, 2020). Briefly, cell lyses were resolved by SDS-PAGE, and the separated proteins were transferred to PVDF membranes (Millipore). The membranes were blocked at room temperature using 5% skim milk (Boster) and then incubated with the corresponding primary and secondary antibodies, respectively. After washing, development was performed using ECL detection reagents (MeilunBio, MA0186), with Tanon 5200 chemiluminescence imaging system and ImageJ (NIH, 1.50i) software to quantify the density of protein bands. Next, relative protein expression was determined by the ratio of the greyscale value of the target protein to that of the internal reference, with normalization relative to the WT controls. The antibodies used in this study are listed in the Reagents and tools table.

### Lipids, lipid peroxidation, and cell survival analysis

The Micro Total Cholesterone (TC) Content Assay Kit and Micro Free Cholesterone (FC) Content Assay Kit (Solarbio, BC1985, BC1895) were used to measure cholesterol and free cholesterol levels; the Modified Oil Red O Stain Kit (Solarbio, G1263) was used to measure lipids; the CheKine™ Micro Triglyceride (TG) Assay Kit (Abbkine, KTB2200) was used to measure TG levels; the CheKine™ Micro Lipid Peroxidation (MDA) Assay kit (Abbkine, KTB1050) was used to measure MDA levels; the free cholesterol was stained using Filipin (MeilunBio, MB1848); the Meilun Reactive Oxygen Species Assay Kit (MeilunBio, MA0219) was used to measure ROS levels; the Cell Counting Kit-8 (CCK8, APExBIO) was used to measure cell survival.

### Intracellular calcium concentration analysis

The Fura-2 AM calcium ion fluorescent probe (MeilunBio, MA0194) was used to analyze the intracellular calcium concentration of the cell lines. Then the dual wavelength detection at 340 and 380 nm was determined using Infinite 200 PRO (Tecan).

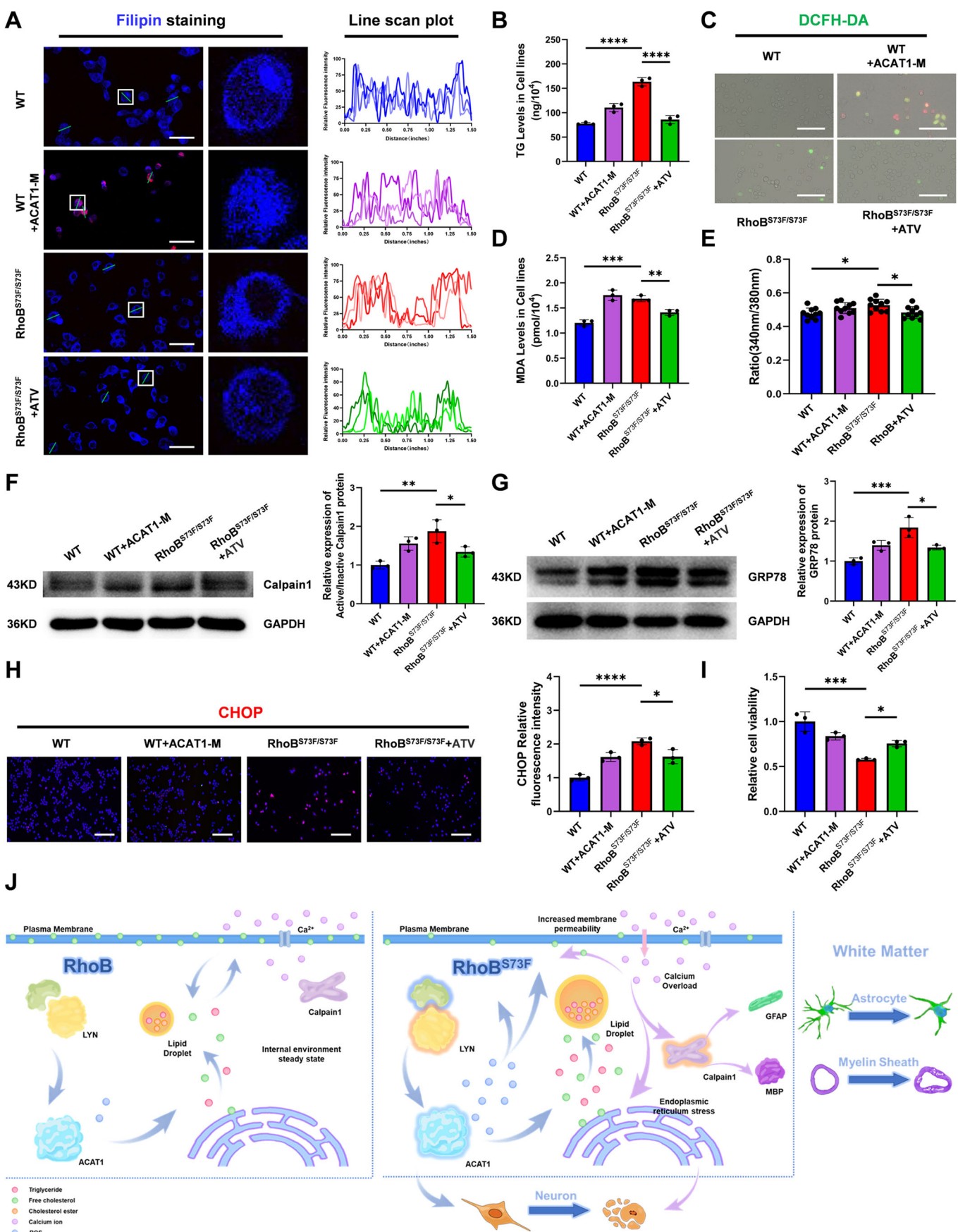

◀ **Figure 6.  Disruption of lipid homeostasis by the RhoB p.S73F mutation could be alleviated by atorvastatin treatment.**

(A) Filipin staining (Blue) images and line scan plot analysis of FC in RhoB$^{S73F/S73F}$ cells ($n = 3$ per group). The significant recovery of FC homeostasis was observed in the inhibition cells (Fisher's exact test; $P = 5.93^{E-04}$). pmCherry-C1-ACAT1-M (Red). Scale bars: 50 μm. (B) TG levels analysis in RhoB$^{S73F/S73F}$ cells ($n = 3$ per group). A significant recovery of TG levels was observed in the inhibition cells. See details on $P$ values in Appendix Table S26. (C) DCFH-DA (Green) levels analysis in RhoB$^{S73F/S73F}$ cells ($n = 3$ per group). A significant recovery of ROS levels was observed in the inhibition cells. pmCherry-C1-ACAT1-M (Red). Scale bars: 100 μm. (D) MDA levels analysis in RhoB$^{S73F/S73F}$ cells ($n = 3$ per group). A significant recovery of MDA levels was observed in the inhibition cells. See details on $P$ values in Appendix Table S27. (E) Intracellular calcium ion levels analysis in RhoB$^{S73F/S73F}$ cells ($n = 10$ per group). The significant restoration of calcium homeostasis was observed in the inhibition cells. See details on $P$ values in Appendix Table S28. (F) WB detection and relative expression analysis of activated Calpain1 in RhoB$^{S73F/S73F}$ cells ($n = 3$ per group). The significantly reduced activated Calpain1 protein levels were observed in the inhibition cells. See details on $P$ values in Appendix Table S29. (G) WB detection and relative expression analysis of GRP78 in RhoB$^{S73F/S73F}$ cells ($n = 3$ per group). The significantly reduced GRP78 protein was observed in the inhibition cells. See details on $P$ values in Appendix Table S30. (H) Immunofluorescence staining images and relative fluorescence intensity analysis of CHOP (Red) in RhoB$^{S73F/S73F}$ cells ($n = 3$ per group). The significantly reduced CHOP protein was observed in the inhibition cells. mEGFP-N1-ACAT1-M (Green); Nuclei DAPI-stained (Blue). Scale bars: 500 μm. See details on $P$ values in Appendix Table S31. (I) Relative cell viability analysis in RhoB$^{S73F/S73F}$ cells ($n = 3$ per group). The significant restoration of relative cell viability was observed in the inhibition cells. See details on $P$ values in Appendix Table S32. (J) Schematic representation of the molecular pathogenesis for dysregulated lipid homeostasis and neurological damage caused by RhoB$^{S73F/S73F}$ mutation via LYN-ACAT1. Data information: Data represent different numbers ($n$) of biological replicates. In (B, D–I), data are presented as mean ± SD (One-way ANOVA). *$P \leq 0.05$, **$P \leq 0.01$, ***$P \leq 0.001$, ****$P \leq 0.0001$. Source data are available online for this figure.

## Statistical analyses

Quantitative and statistical analyses were conducted using Graph-Pad Prism 8. This study included a minimum of three independent biological replicates for each experiment, with results presented as mean ± standard deviation (SD). All experiments were conducted in a randomized manner and analyzed in a blinded fashion. All available samples were included in the analyses without any specific inclusion/exclusion criteria. Unpaired two-tailed Student's $t$-tests were utilized for comparing two groups, while one-way ANOVA was employed for comparing more than two groups. Fisher's exact test was applied to assess the effect of ATV on the restoration of FC homeostasis in RhoB$^{S73F/S73F}$ cells. A significance level of $P < 0.05$ was considered statistically significant for all analyses.

## For More Information

https://www.rcsb.org/

## Data availability

Microscopic images and mass spectrometry data can be obtained from the BioImage Archive under accession number S-BIAD1163.

The source data of this paper are collected in the following database record: biostudies:S-SCDT-10_1038-S44321-024-00113-2.

## Peer review information

## The paper explained

### Problem

Cerebral palsy (CP) is a prevalent neurological disorder. A recent discovery has connected the RhoB p.S73F mutation to CP, but the precise pathogenesis resulting from this mutation remains uncertain. Furthermore, current animal models of CP do not show signs of spasticity and motor deficits and do not have any association with known cases of genetic CP.

### Results

A rabbit model with the RhoB p.S73F mutation was established, which successfully replicated the progression of human CP, exhibiting typical symptoms, such as periventricular leukomalacia and spastic-dystonic paraplegia. Additionally, findings from animal models may suggest that excessive activation of ACAT1, caused by stronger binding affinity of the RhoB p.S73F protein to LYN of the RhoB p.S73F protein to LYN, could disrupt lipid homeostasis, leading to lipid peroxidation, calcium overload, and subsequent damage to neuronal cells and myelin.

### Impact

This study presented the first mammalian model of genetic CP that accurately replicates the RhoB p.S73F mutation in humans, provided further insights between RhoB and lipid metabolism, and novel therapeutic targets for human CP.

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

## Acknowledgements

This study was financially supported by the National Natural Science Foundation of China (32170543), the Young Elite Scientists Sponsorship Program by CAST (2022QNRC001), and the Natural Science Foundation of Jilin Province of China (20230101154JC). The authors thank Peiran Hu and Nannan Li for assistance at the Embryo Engineering Center for the critical technical assistance.

## Author contributions

**Xinyu Wu**: Data curation; Formal analysis; Validation; Investigation; Visualization; Methodology; Writing—original draft; Writing—review and editing. **Ruonan Liu**: Data curation; Formal analysis; Validation; Investigation; Visualization; Methodology; Writing—original draft; Writing—review and editing. **Zhongtian Zhang**: Formal analysis; Visualization; Methodology. **Jie Yang**: Formal analysis; Investigation; Visualization. **Xin Liu**: Formal analysis; Investigation; Visualization. **Liqiang Jiang**: Formal analysis; Investigation; Visualization. **Mengmeng Fang**: Formal analysis; Investigation; Visualization. **Shoutang Wang**: Visualization; Methodology; Writing—review and editing. **Liangxue Lai**: Conceptualization; Resources; Supervision; Funding acquisition; Project administration; Writing—review and editing. **Yuning Song**: Conceptualization; Resources; Supervision; Funding acquisition; Writing—original draft; Project administration; Writing—review and editing. **Zhanjun Li**: Conceptualization; Resources; Supervision; Funding acquisition; Project administration; Writing—review and editing.

Source data underlying figure panels in this paper may have individual authorship assigned. Where available, figure panel/source data authorship is listed in the following database record: biostudies:S-SCDT-10_1038-S44321-024-00113-2.

## Disclosure and competing interests statement

The authors declare no competing interests.

# Expanded View Figures

**Figure EV1.**  (A) Summary of mutation efficiency and development of embryos after injection with the CRISPR/Cas9 system ($n = 30$ per group). (B) Sanger sequencing peak map of F0 generation rabbit ear clips. The red box indicates the site of the mutation. (C) Sanger sequencing results of RhoB gene mutation in newborn gene editing rabbits. Red denotes a mutant base; green denotes a PAM sequence; the underline indicates the design location of the SgRNA. (D) Summary of F1 generation rabbit breeding situation. F1 generation rabbits with a stable inheritance of the RhoB mutation were obtained. (E) Body weight curves of age-matched WT controls ($n = 3$) and RhoB$^{S73F/+}$ rabbits ($n = 3$). A significantly reduced body weight was observed in RhoB$^{S73F/+}$ rabbits. (F) Survival curves of age-matched WT controls ($n = 4$) and RhoB$^{S73F/+}$ rabbits ($n = 8$). A significantly reduced survival rate was observed in RhoB$^{S73F/+}$ rabbits. (G) Representative postural images of 12-week-old WT controls ($n = 3$) and RhoB$^{S73F/+}$ rabbits ($n = 3$). Severe motor and postural control issues were observed in RhoB$^{S73F/+}$ rabbits. (H, I) Representative gait trajectories (H) and gait analysis (H) of 12-week-old WT controls ($n = 5$) and RhoB$^{S73F/+}$ rabbits ($n = 5$). The significantly varied pace trajectory was observed in RhoB$^{S73F/+}$ rabbits. Red indicates front paw tracks; blue indicates hind paw tracks. The dataset includes measurements of stride length, step width, and step length of the hind limb footprint of rabbits during movement. See details on $P$ values in Appendix Table S33. Data information: Data represent different numbers ($n$) of biological replicates. In (I), data were presented as mean ± SD (Unpaired two-tailed Student's $t$-tests). ****$P \leq 0.0001$. Source data are available online for this figure.

**A**

|  | Zygotes, n | Two-cell, n (%) | Morula, n (%) | Blastocyst, n (%) | Mutant blastocyst, n (%) |
|---|---|---|---|---|---|
| Noninjection | 30 | 29 (96.7) | 28 (93.3) | 26 (86.7) | 0 |
| Injection | 30 | 28 (93.3) | 27 (90) | 25 (83.3) | 25 (83.3) |

**B**

**C**

```
WT      GACCGCCTGCGGCCGCTCTCCTACCCGGACACCGACGTCATCCTCATG
#F0-1   GACCGCCTGCGGCCGCTCTTTTACCCGGACACCGACGTCATCCTCATG
#F0-2   GACCGCCTGCGGCCGCTCTCCTACCCGGACACCGACGTCATCCTCATG
        GACCGCCTGCGGCCGCTCTGTTACCCGGACACCGACGTCATCCTCATG
#F0-3   GACCGCCTGCGGCCGCTCTCCTACCCGGACACCGACGTCATCCTCATG
        GACCGCCTGCGGCCGCT-TAATACCCGGACACCGACGTCATCCTCATG
#F0-4   GACCGCCTGCGGCCGCTCTCCTACCCGGACACCGACGTCATCCTCATG
        GACCGCCTGCGGCCGCTCTGTTACCCGGACACCGACGTCATCCTCATG
#F0-5   GACCGCCTGCGGCCGCTCTCCTACCCGGACACCGACGTCATCCTCATG
        GACACC--------------------GACACCGACGTCATCCTCATG
#F0-6   GACCGCCTGCGGCCGCTCTCCTACCCGGACACCGACGTCATCCTCATG
        GACCGCCTGCGGCCGCTCTTTACCCGGACACCGACGTCATCCTCATG
#F0-7   GACCGCCTGCGGCCGCTCTCCTACCCGGACACCGACGTCATCCTCATG
        GACCGCCTGCGGCCGCTCTTTTACCCGGACACCGACGTCATCCTCATG
#F0-8   GACCGCCTGCGGCCGCTCTCCTACCCGGACACCGACGTCATCCTCATG
        GACCGCCTGCGGCCGCTCTTTACCCGGACACCGACGTCATCCTCATG
#F0-9   GACCGCCTGCGGCCGCTCTTTTACCCGGACACCGACGTCATCCTCATG
#F0-10  GACCGCCTGCGGCCGCTCTTTACCCGGACACCGACGTCATCCTCATG
#F0-11  GACCGCCTGCGGCCGCTCTCCTACCCGGACACCGACGTCATCCTCATG
        GACCGCCTGCGGCCGCTCTTTTACCCGGACACCGACGTCATCCTCATG
```

**D**

```
GACCGCCTGCGGCCGCTCTTTTACCCGGACACCGACGTCATCCTCATG  (S73F)  X6
GACCGCCTGCGGCCGCTCTTTTACCCGGACACCGACGTCATCCTCATG  (S73F)

GACCGCCTGCGGCCGCTCTTTTACCCGGACACCGACGTCATCCTCATG  (S73F)  X8
GACCGCCTGCGGCCGCTCTCCTACCCGGACACCGACGTCATCCTCATG  (WT)

GACCGCCTGCGGCCGCTCTCCTACCCGGACACCGACGTCATCCTCATG  (WT)   X4
GACCGCCTGCGGCCGCTCTCCTACCCGGACACCGACGTCATCCTCATG  (WT)
```

**E**

**F**

**G**

**H**

**I**

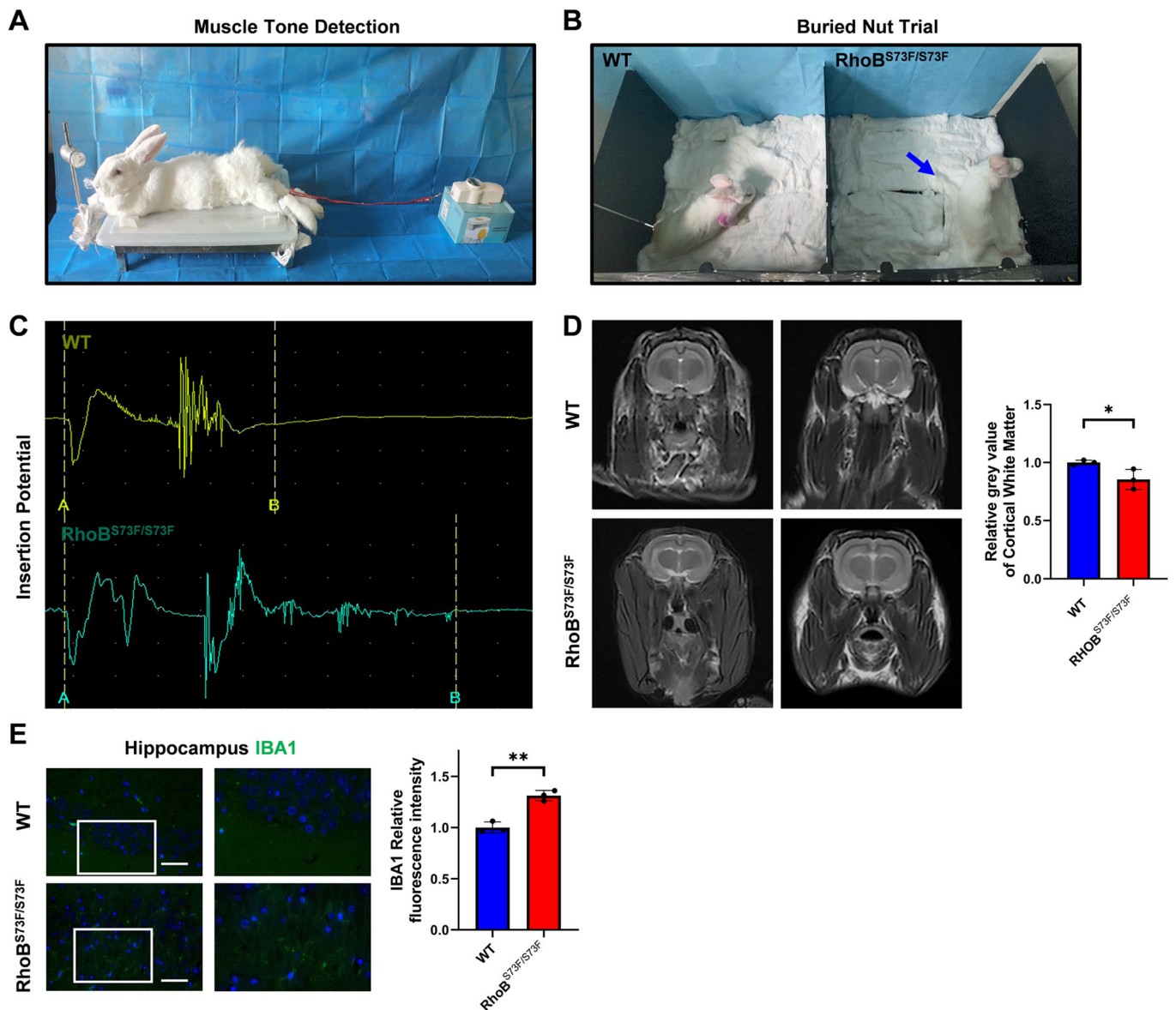

**Figure EV2.** (A) Representative schematic of lower limb muscle tone testing in rabbits. (B) Representative schematic of the buried nut trials in rabbits.
(C) Representative needle electromyography images of insertion potential in the left hind limb gastrocnemius muscle from 12-week-old WT controls (*n* = 3) and
RhoB^S73F/S73F rabbits (*n* = 3). The significant prolongation of insertion potential was observed in RhoB^S73F/S73F rabbits. Myoelectric Filters: 10 to 5 kHz; Scanning rate and
sensitivity: 100 ms/div, 5 mV/div. Scale bars: 50 ms; 2.5 mV. (D) MRI detection images and relative gray value analysis of white matter in the brain from 12-week-old
WT controls (*n* = 3) and RhoB^S73F/S73F rabbits (*n* = 3). Reduced brain white matter density was observed in RhoB^S73F/S73F rabbits. See details on *P* values in Appendix
Table S34. (E) Immunofluorescence staining images and relative fluorescence intensity analysis of IBA1 (Green) in brain sections from 12-week-old WT controls (*n* = 3)
and RhoB^S73F/S73F rabbits (*n* = 3). The significantly activated microglia were observed in RhoB^S73F/S73F rabbits. Nuclei DAPI-stained (Blue). Scale bars: 100 μm. See details on
*P* values in Appendix Table S35. Data information: Data represent different numbers (*n*) of biological replicates. In (D, E), data were presented as mean ± SD (Unpaired
two-tailed Student's *t*-tests). *$P \leq 0.05$, **$P \leq 0.01$. Source data are available online for this figure.

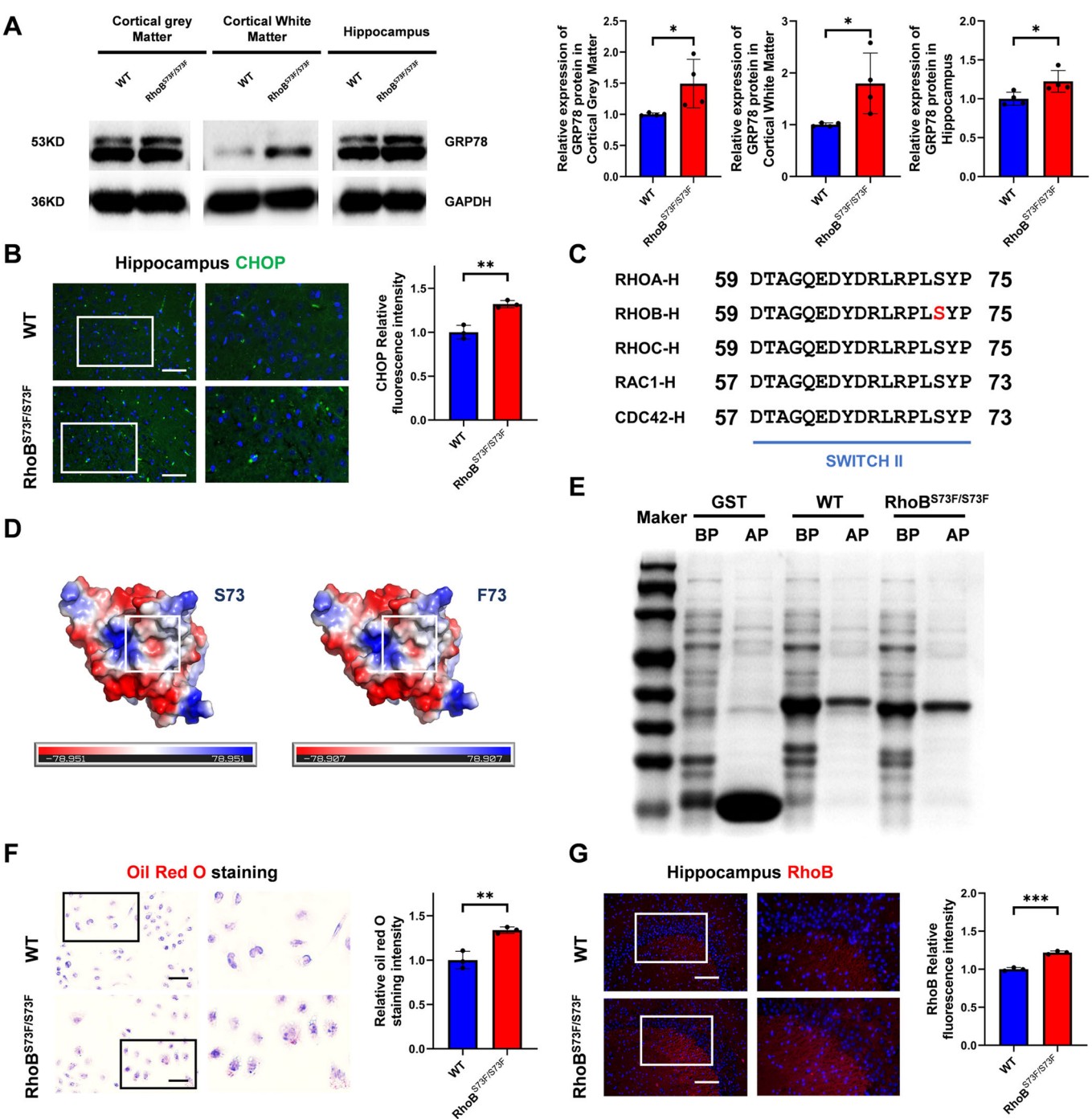

**Figure EV3.** (**A**) WB detection and relative expression analysis of GRP78 in the brain from 12-week-old WT controls (*n* = 4) and RhoB^S73F/S73F^ rabbits (*n* = 4). The significantly upregulated GRP78 protein was observed in RhoB^S73F/S73F^ rabbits. See details on *P* values in Appendix Table S36. (**B**) Immunofluorescence staining images and relative fluorescence intensity analysis of CHOP (Green) in brain sections from 12-week-old WT controls (*n* = 3) and RhoB^S73F/S73F^ rabbits (*n* = 3). The significantly increased CHOP protein was observed in RhoB^S73F/S73F^ rabbits. Nuclei DAPI-stained (Blue). Scale bars: 100 μm. See details on *P* values in Appendix Table S37. (**C**) Homologous family sequence alignment of the SWITCH II domain of RhoB protein. The Switch II structural domain of the RhoB gene is highly conserved within the homologous family. The mutant residue is showed in red. (**D**) Comparison of the surface potential of WT control and p.S73F mutant group of human-derived RhoB protein. The local surface potential change was observed in the RhoB^S73F/S73F^ protein. Data from RCSB Protein Data Bank. (**E**) SDS-PAGE analysis for the purification of GST, WT, and RhoB^S73F/S73F^ recombinant protein. GST indicates GST protein control; WT indicates GST-RhoB recombinant protein; RhoB^S73F/S73F^ indicates GST-RhoBS73F/S73F mutant protein; BP indicates before purification; AP indicates after purification. (**F**) Oil red O staining images and relative intensity (OD value) analysis in WT controls (*n* = 3) and RhoB^S73F/S73F^ cells (*n* = 3). Significant lipid droplet aggregation was observed in the RhoB^S73F/S73F^ cells. Scale bars: 50 μm. See details on *P* values in Appendix Table S38. (**G**) Immunofluorescence staining images and relative fluorescence intensity analysis of RhoB (Red) in brain sections from 12-week-old WT controls (*n* = 3) and RhoB^S73F/S73F^ rabbits (*n* = 3). The significantly upregulated RhoB protein was observed in RhoB^S73F/S73F^ rabbits compared to the WT controls. Nuclei DAPI-stained (Blue). Scale bars: 200 μm. See details on *P* values in Appendix Table S39. Data information: Data represent different numbers (*n*) of biological replicates. In (**A, B, F, G**), data were presented as mean ± SD (Unpaired two-tailed Student's *t*-tests). **P* ≤ 0.05, ***P* ≤ 0.01, ****P* ≤ 0.001. Source data are available online for this figure.

