## [Peer Review File · EMBO Molecular Medicine]

The RhoB p.S73F mutation leads to cerebral palsy through dysregulation of lipid homeostasis

Zhanjun Li, Xinyu Wu, Ruonan Liu, Zhongtian Zhang, Jie Yang, Xin Liu, Liqiang Jiang, Mengmeng Fang, Shoutang Wang, Liangxue Lai, and Yuning Song

Corresponding authors: Zhanjun Li (lizj_1998@jlu.edu.cn), Liangxue Lai (lai_liangxue@gibh.ac.cn), Yuning Song (songyuning0313@jlu.edu.cn)

Review Timeline:

Submission Date:	22nd Dec 23
Editorial Decision:	13th Feb 24
Revision Received:	13th May 24
Editorial Decision:	18th Jun 24
Revision Received:	9th Jul 24
Accepted:	15th Jul 24

Editor: Zeljko Durdevic

Transaction Report:

13th Feb 2024

Dear Prof. Li,

Thank you for the submission of your manuscript to EMBO Molecular Medicine. We have now received feedback from the three reviewers who agreed to evaluate your manuscript. All three referees recognize potential interest of the study but also raise important and partially overlapping criticism that should be addressed in a major revision. I would also recommend running the article by a native English speaker. If you would like to discuss further the points raised by the referees, I am available to do so via email or video. Let me know if you are interested in this option.

We would welcome the submission of a revised version within three months for further consideration. Please let us know if you require longer to complete the revision.

I look forward to receiving your revised manuscript.

Yours sincerely,

Zeljko Durdevic

We require:

- 1) A .docx formatted version of the manuscript text (including legends for main figures, EV figures and tables). Please make sure that the changes are highlighted to be clearly visible.
- 2) Individual production quality figure files as .eps, .tif, .jpg (one file per figure). For guidance, download the 'Figure Guide PDF': (<https://www.embopress.org/page/journal/17574684/authorguide#figureformat>).
- 3) A .docx formatted letter INCLUDING the reviewers' reports and your detailed point-by-point responses to their comments. As part of the EMBO Press transparent editorial process, the point-by-point response is part of the Review Process File (RPF), which will be published alongside your paper.
- 4) A complete author checklist, which you can download from our author guidelines (<https://www.embopress.org/page/journal/17574684/authorguide#submissionofrevisions>). Please insert information in the checklist that is also reflected in the manuscript. The completed author checklist will also be part of the RPF.
- 5) Please note that all corresponding authors are required to supply an ORCID ID for their name upon submission of a revised manuscript.

6) It is mandatory to include a 'Data Availability' section after the Materials and Methods. Before submitting your revision, primary datasets produced in this study need to be deposited in an appropriate public database, and the accession numbers and database listed under 'Data Availability'. Please remember to provide a reviewer password if the datasets are not yet public (see <https://www.embopress.org/page/journal/17574684/authorguide#dataavailability>).

13) Author contributions: You will be asked to provide CRediT (Contributor Role Taxonomy) terms in the submission system. These replace a narrative author contribution section in the manuscript.

14) A Conflict of Interest statement should be provided in the main text.

Please also suggest a striking image or visual abstract to illustrate your article as a PNG file 550 px wide x 300-800 px high.

**** Reviewer's comments ****

Referee #1 (Comments on Novelty/Model System for Author):

Two identical de novo missense variants, p.Ser73Phe, in RHOB have been identified in cerebral palsy (CP) cases, providing strong evidence for RHOB as a disease-causing gene in CP. The authors have generated a knock-in rabbit model with the same RHOB variant and found that the mutant rabbit could recapitulate the phenotype of human CP. The authors also showed that the mutant RHOB could lead to brain lesions at tissue and cellular level. Finally, that authors provided in vitro experimental data demonstrating that the effect of the mutant RHOB could be inverted by Atorvastatin. The manuscript is very interesting, confirming the role of the RHOB gene in the genetic etiology of human CP and providing a therapeutic target for human CP. My concern is the language. The English need be polished!

Referee #1 (Remarks for Author):

The authors have generated a knock-in rabbit model with the same RHOB variant and found that the mutant rabbit could recapitulate the phenotype of human CP. The authors also showed that the mutant RHOB could lead to brain lesions at tissue and cellular level. Finally, that authors provided in vitro experimental data demonstrating that the effect of the mutant RHOB could be inverted by Atorvastatin. The manuscript is very interesting, confirming the role of the RHOB gene in the genetic etiology of human CP and providing a therapeutic target for human CP.

Referee #2 (Remarks for Author):

The underlying genetic underpinnings that contribute to cerebral palsy (CP) remain largely unknown. A recent study showed that the RHOB p.S73F mutation is associated with cerebral palsy, but how this mutation contributes to CP pathology is unclear. In this research article, the authors focus on investigating the molecular mechanisms by which the RHOB p.S73F mutation contributes to CP pathology. In this elegant study, the authors generate a novel rabbit model containing the RHOB p.S73F mutation using the sgRNA-directed CRISPR/Cas9 SpG-BE4max system. They then show the mutant rabbit model shows the classical signs of CP including periventricular leukomalacia and spastic-dystonic diplegia. Using a variety of molecular and biochemical analyses, the authors demonstrate a novel finding that the RHOB p.S73F mutation causes dysregulation of lipid homeostasis through specific activation of ACAT1 through the LYN pathway. The authors implement well-designed experiments with proper controls and overall the article is written well. While my enthusiasm for this article remain high, I do have a significant number of minor reservations specific to grammar/word choice and interpretation of data that if addressed would provide for an excellent contribution to EMBO Molecular Medicine.

Minor Concerns:

- 1.) The title of the article can be re-written for clarity. I would suggest "The RHOB S73F mutation leads to cerebral palsy through dysregulation of lipid homeostasis"
- 2.) In line 62, please re-word "Moreover, based on the model rabbits" to "Moreover, based on the rabbit model of CP"
- 3.) In line 77, it is unclear what the meaning of this phrase refers to: "15% of embryos were edited with 100% efficiency"
- 4.) In line 93, I would suggest being more concise in the timing of death. Fig 1D suggests the mutant rabbits died within 26 weeks.
- 5.) In line 157, since there is a significant increase of RHOB protein in the RHOB S73F mutant brain, I would suggest rewording this sentence: "of interest showed an upregulation trend in the mutant rabbit brain" to "of interest showed a significant increase in the mutant rabbit brain"
- 6.) In line 195, I would suggest changing "binding ability" to "binding affinity"
- 7.) In line 200, I would suggest changing "most significant differential protein ACAT1" to "most significant differentially bound protein ACAT1"

- 8.) In line 206, please reword "Moreover" to "Also"
- 9.) In line 222, I would suggest rewording "displayed cholesterol disorders" to "displayed disruption in cholesterol localization"
- 10.) It would strengthen the analysis of ectopic free cholesterol in control and mutant cell lines in Fig. 6A by quantifying these results.
- 11.) In line 278, "Pull-Down" should be written as "pull-down"
- 12.) In Fig.3K, it would be useful to label the X-axis as "Perivascular GFAP Relative IHC Intensity"
- 13.) I would suggest to reword Figure 4 as "Lipid homeostasis is disrupted in RHOB S73F/ S73F rabbits."
- 14.) Results from Figure 5B and within the figure legend and text descriptions should be referred to as change in relation to differential binding or fold change in abundance of binding and not to upregulated or downregulated.

Referee #3 (Remarks for Author):

The authors present an exciting new rabbit model for genetic cerebral palsy (CP) based on recent genomic discoveries. This is a first-in-kind model with mechanistic insights that may be highly significant as the group identifies molecular and cellular pathology that may be contributing to an important genetic form of cerebral palsy. To date, models for genetic forms of cerebral palsy have been sparse, so this work may represent a major advance for the field. Additionally, many of the mechanisms uncovered here may be relevant to other genetic and acquired forms of cerebral palsy and could have widespread positive impact for the understanding underlying of pathology.

The authors present a detailed characterization of this model and make the case that the homozygous RHOB F73S mutation in rabbits leads to a CP-like phenotype with core features recapitulating the features seen in human patients. After extensive phenotyping studies, they present diverse functional experiments to identify changes in binding affinity and activation of ACAT1 via LYN associated with lipid abnormalities.

However, enthusiasm for the paper in its current form is diminished due to potential overstatements of some conclusions, failure to provide quantitative data to substantiate some claims, some lacking details (including methods not clearly documented in accordance with the policies outlined in the journal's guide to authors). If authors could address these concerns, enthusiasm for the paper would increase.

Major concerns

- The human p.F73S RHOB variant occurs in heterozygous form, yet the animal studies are homozygous. This may be acceptable, but needs to be transparently addressed.
- The RHOB animal model seems to exhibit peripheral neuropathy in addition to brain abnormalities. This would not preclude a diagnosis of CP in humans, but is not "run of the mill." This should be mentioned. It is an interesting finding however that may be worth assessing in human patients with RHOB
- The use of the term "dyskinesia" to describe the rabbits' reduced movement distance, increased righting time, etc is not consistent with contemporary usage in humans and appears incongruent (semantic but important for a manuscript that very clearly aims to translate findings across systems). The term "hypertonia" may be applicable in some contexts or the more general term "neuromotor impairment" may be appropriate.
- Throughout the paper, individual data points need to be represented on plots consistently and reported in the figure legends. Fishers exact test may be more appropriate in some instances given assumptions about variability and very low sample sizes. Animal ages should be reported in figure legends. Some context about what is being measured and how should also be in the figure legend (ex. What is measured in muscle fiber grouping? What do the colors shown in the panel represent?) Panel labels often hard to see due to low contrast). What features are being described as abnormal in baseline EMG? Variability? Frequency? Quantification, labeling features, and figure legend descriptions could be improved throughout.

Minor concerns

- Introduction - The first signs of CP are often recognizable by age 6 months, sometimes earlier, prompting emphasis on early diagnosis (PMID: 28715518).
- The role of RHOB in cancer should mention that such mutations are typically somatic rather than germline.
- References 25-27 do not seem to describe state-of-the-science in cerebral palsy genomics
- Line 65: Although this may be one of the first mammalian models of genetic CP, it is not one of the first to model CP overall
- How was muscle tone in the rabbits assessed? This represents an important detail in a new model
- MRI data does not provide information about who interpreted the imaging, what imaging parameters were used, how many animals were studied, at what age, etc. Figure legend does not clearly label what is seen in each image. No measurements of white matter tract sizes (and comparison to control animals) were provided.
- More information on immunohistochemistry methods should be provided. What equipment was used, what software, how many fields of view imaged and how were they selected? What image processing was used before quantification of signal intensity? Was there blinding to condition for processing or scoring? Supplemental table with antibodies used and concentrations missing. How was reduced neuron number determined?
- The authors should qualify their statements linking calpain, dysregulated calcium signaling, and ER stress to avoid

overstatement. There are alternative interpretations, particularly in the absence of independent experiments and methodological detail.

- Information on animal use (adherence to ARRIVE guidelines) and ethical committee oversight could not be located.
- The last paragraph of the Results section should be qualified to indicate that the events discussed "could" be connected.
- Lipid droplets are not clearly shown (i.e. labeling with arrows described in figure legends or with an independent staining marker) nor quantified. The conclusion that RHOB73F mutations increase lipid droplet accumulation should be better substantiated.

**Responses to Reviewer's Comments:**

Thank you very much for your comments concerning our manuscript entitled "The
RhoB p.S73F mutation leads to cerebral palsy through dysregulation of lipid
homeostasis (EMM-2023-19193-T)". Those comments are all valuable and very
helpful for revising and improving our paper, as well as the important guiding
significance to our researches. We have studied comments carefully and made
correction which we hope meet with approval. Revised portions are marked with **blue**
**fonts** in the "revised highlighted" copy. The main corrections in the paper and the
responses to the comments are as follows.

**To Editor:**

**Response:** Thank you for considering this manuscript. We have carefully reviewed
the instructions and suggestions of editor, and experimental details, including
histological image acquisition methods, data analysis methods, and animal use
guidelines, have been added in revised manuscript accordingly.

**To Reviewer #1:**

1.* My concern is the language. The English needs to be polished!

**Response:** Thank you for your important suggestion. The manuscript has been read
carefully, and the English writing has been modified by a native English speaker and a
professional paper writing agency (American Journal Experts, Verification code:
32AB-6D94-9D38-90AD-95B5) accordingly. All the changes are highlighted in blue.

**To Reviewer #2:**

1.* The title of the article can be re-written for clarity. I would suggest "The RHOB
S73F mutation leads to cerebral palsy through dysregulation of lipid homeostasis"

**Response:** Thank you for your kind suggestion. The title of the manuscript has been
changed in the revised manuscript.

2.* In line 62, please re-word "Moreover, based on the model rabbits" to "Moreover,
based on the rabbit model of CP"

**Response:** Thank you for your kind suggestion. The description has been changed in
line 77 of the revised manuscript.

**3.*** In line 77, it is unclear what the meaning of this phrase refers to: "15% of embryos
were edited with 100% efficiency"

**Response:** Thank you for your good suggestion. The description has been changed in
line 89 of the revised manuscript.

**4.*** In line 93, I would suggest being more concise in the timing of death. Fig 1D
suggests the mutant rabbits died within 26 weeks.

**Response:** Thank you for the helpful comment. The description has been changed in
line 100 of the revised manuscript.

**5.*** In line 157, since there is a significant increase of RHOB protein in the RHOB
S73F mutant brain, I would suggest rewording this sentence: "of interest showed an
upregulation trend in the mutant rabbit brain" to "of interest showed a significant
increase in the mutant rabbit brain"

**Response:** Thank you for your kind suggestion. The description has been changed in
line 259 of the revised manuscript.

**6.*** In line 195, I would suggest changing "binding ability" to "binding affinity"

**Response:** Thank you for your valuable suggestion. The description has been changed
in line 174 of the revised manuscript.

**7.*** In line 200, I would suggest changing "most significant differential protein
ACAT1" to "most significant differentially bound protein ACAT1"

**Response:** Thank you for your valuable feedback. The description has been changed
in line 176 of the revised manuscript.

**8.*** In line 206, please reword "Moreover" to "Also"

**Response:** Thank you for your good suggestion. The description has been changed in
line 182 of the revised manuscript.

**9.*** In line 222, I would suggest rewording "displayed cholesterol disorders" to
"displayed disruption in cholesterol localization"

**Response:** Thank you for your valuable suggestion. The description has been changed
in line 199 of the revised manuscript.

**10.*** It would strengthen the analysis of ectopic free cholesterol in control and mutant
cell lines in Fig. 6A by quantifying these results.

**Response:** Thank you for your insightful feedback. The localisation of free
cholesterol has been analysed using line scan plots and the ectopic free cholesterol has
been quantified in Fig. 6A of the revised manuscript.

**11.*** In line 278, "Pull-Down" should be written as "pull-down"

**Response:** Thank you for your kind suggestion. The description has been changed in
line 254 of the revised manuscript.

**12.*** In Fig.3K, it would be useful to label the X-axis as "Perivascular GFAP Relative
IHC Intensity"

**Response:** Thank you for your insightful feedback. The X-axis label has been
changed in Fig. 3H of the revised manuscript.

**13.*** I would suggest to reword Figure 4 as "Lipid homeostasis is disrupted in RHOB
S73F/ S73F rabbits.

**Response:** Thank you for your valuable comments. The title of Fig. 4 has been
changed in line 726 of the revised manuscript.

**14.*** Results from Figure 5B and within the figure legend and text descriptions should
be referred to as change in relation to differential binding or fold change in abundance
of binding and not to upregulated or downregulated.

**Response:** Thank you for your kind suggestion. The figure legend and text
descriptions of Fig. 5B have been changed in line 753 of the revised manuscript,
respectively.

**To Reviewer #3:**

**Referee #3 (Remarks for Author):**

However, enthusiasm for the paper in its current form is diminished due to potential
overstatements of some conclusions, failure to provide quantitative data to
substantiate some claims, some lacking details (including methods not clearly
documented in accordance with the policies outlined in the journal's guide to authors).

**Response:** Thank you for your kind suggestion.

The overstatements in line 28, 213 and 274 have been changed in the revised
manuscript, respectively.

The quantitative data on muscle tone, brain white matter, neuron number,
intracellular free cholesterol, and intracellular lipid droplets have been included in
Fig. 1J, EV2D, 3F, 6A, and EV3F of the revised manuscript.

The lacking details, including histological image acquisition methods, data
analysis methods, and animal use guidelines, have been included in line 360, 418, and
280 of the revised manuscript.

**Major concerns**

**1.*** The human p.F73S RHOB variant occurs in heterozygous form, yet the animal
studies are homozygous. This may be acceptable, but needs to be transparently
addressed.

**Response:** Thank you for your kind suggestion. In studies involving CP model
animals, it has been observed that RhoB^{S73F/S73F} rabbits typically develop the disease
between 2 and 4 weeks of age, with a faster disease progression. On the other hand,
RhoB^{S73F/+} rabbits may develop the disease between 2 weeks of age and 8 months of
age, with a slower disease progression. Despite all rabbits exhibiting a highly
consistent spastic paraplegia phenotype in the final stage of the disease process, the
onset of the disease is more stable and the phenotype more pronounced in
RhoB^{S73F/S73F} rabbits. Therefore, RhoB^{S73F/S73F} rabbits were chosen as the primary
subjects for this study. Corresponding content has been included in lines 96 and Fig.
EV1E-EVII of the revised manuscript.

**2.*** The RHOB animal model seems to exhibit peripheral neuropathy in addition to
brain abnormalities. This would not preclude a diagnosis of CP in humans, but is not
"run of the mill." This should be mentioned. It is an interesting finding however that
may be worth assessing in human patients with RHOB

**Response:** Thank you for your insightful feedback. The peripheral neuropathy
phenotype of common peroneal nerve has been discussed in line 238 of the revised
manuscript.

**3.*** The use of the term "dyskinesia" to describe the rabbits' reduced movement
distance, increased righting time, etc is not consistent with contemporary usage in
humans and appears incongruent (semantic but important for a manuscript that very
clearly aims to translate findings across systems). The term "hypertonia" may be

applicable in some contexts or the more general term "neuromotor impairment" may
be appropriate.

**Response:** Thank you for your kind suggestion. The description has been changed in
line 76, 97, 102, 111 and 225 of the revised manuscript.

**4.*** Throughout the paper, individual data points need to be represented on plots
consistently and reported in the figure legends. Fishers exact test may be more
appropriate in some instances given assumptions about variability and very low
sample sizes. Animal ages should be reported in figure legends. Some context about
what is being measured and how should also be in the figure legend (ex. What is
measured in muscle fiber grouping? What do the colors shown in the panel
represent?) Panel labels often hard to see due to low contrast). What features are being
described as abnormal in baseline EMG? Variability? Frequency? Quantification,
labeling features, and figure legend descriptions could be improved throughout.

**Response:** Thank you very much for the helpful comment.

The figures and figure legends, including the representation and reporting of data
points, animal ages, contents and methods of measurement, and panel labels, have
been changed based on your suggestion in all figures and figure legends of the revised
manuscript.

The effect of atorvastatin on restoring FC homeostasis in the RhoB^{S73F/S73F} cell
line was evaluated using the Fishers exact test, as recommended. The description has
been included in line 424 and 786 of the revised manuscript.

For EMG resting potential detection, high-frequency features have been
highlighted to show discharge abnormalities. The description has been changed in line
122 of the revised manuscript.

**Minor concerns**

**5.*** Introduction - The first signs of CP are often recognizable by age 6 months,
sometimes earlier, prompting emphasis on early diagnosis (PMID: 28715518).

**Response:** Thank you for your good suggestion. The description has been changed in
line 40 of the revised manuscript. The respective literature has been cited.

**6.*** The role of RHOB in cancer should mention that such mutations are typically

somatic rather than germline.

**Response:** Thank you for your kind suggestion. The description has been changed in
line 52 of the revised manuscript.

7.* References 25-27 do not seem to describe state-of-the-science in cerebral palsy
genomics

**Response:** Thank you for your kind suggestion. This sentence summarized the gene-
edited models showed a spasticity phenotype similar to CP, but the genetic changes
that led to this phenotype were not associated with any known cases of CP in the
previous study. The description has been changed in line 64 of the revised manuscript.

8.* Line 65: Although this may be one of the first mammalian models of genetic CP, it
is not one of the first to model CP overall

**Response:** Thank you for your kind suggestion. The description has been changed in
line 79 of the revised manuscript.

9.* How was muscle tone in the rabbits assessed? This represents an important detail
in a new model

**Response:** Thank you for your kind suggestion. Accordingly, the data of muscle tone
in the rabbit models have been assessed and added to Fig. 1J of the revised
manuscript.

10.* MRI data does not provide information about who interpreted the imaging, what
imaging parameters were used, how many animals were studied, at what age, etc.
Figure legend does not clearly label what is seen in each image. No measurements of
white matter tract sizes (and comparison to control animals) were provided.

**Response:** Thank you for your good suggestion. The detail information regarding
MRI-related data, animals, figure legends, and measurements of white matter bundles
have been included in line 324, 690, and Fig. EV2D of the revised manuscript.

11.* More information on immunohistochemistry methods should be provided. What
equipment was used, what software, how many fields of view imaged and how were
they selected? What image processing was used before quantification of signal
intensity? Was there blinding to condition for processing or scoring? Supplemental
table with antibodies used and concentrations missing. How was reduced neuron
number determined?

**Response:** Thank you for your kind suggestion.

More information on immunohistochemistry methods have been added in line
361 of the revised manuscript.

Supplemental table with antibodies used and concentrations missing have been
added in the Reagents and tools table of the revised manuscript.

Reduced neuron number has been quantified and added in line 143 and Fig. 3F of
the revised manuscript.

**12.*** The authors should qualify their statements linking calpain, dysregulated calcium
signaling, and ER stress to avoid overstatement. There are alternative interpretations,
particularly in the absence of independent experiments and methodological detail.

**Response:** Thank you for your kind suggestion. The statement has been qualified.
The description has been changed in line 165 of the revised manuscript.

**13.*** Information on animal use (adherence to ARRIVE guidelines) and ethical
committee oversight could not be located.

**Response:** Thank you very much for your valuable comments. The information on
animal use and ethical committee oversight have been included in line 280 of the
revised manuscript.

**14.*** The last paragraph of the Results section should be qualified to indicate that the
events discussed "could" be connected.

**Response:** Thank you for your kind suggestion. The description has been changed in
line 212 of the revised manuscript.

**15.*** Lipid droplets are not clearly shown (i.e. labeling with arrows described in in
figure legends or with an independent staining marker) nor quantified. The conclusion
that RHOB73F mutations increase lipid droplet accumulation should be better
substantiated.

**Response:** Thank you for your valuable suggestion.

Enhanced labeling of lipid droplets in electron microscopy images has been
included in Fig. 4A of the revised manuscript.

The intracellular lipid droplets have been independently observed and quantified
through oil red O staining. The results have been included in line 194 and Fig. EV3F
of the revised manuscript.

Thank you very much for your email and kindly suggestion. We are glad to receive
further information about the manuscript. We would try our best to edit the
manuscript to fit for the requirement of EMBO Molecular Medicine.

My contact information is: E-mail: lizj_1998@jlu.edu.cn; Tel: (86) 431-87836175;

Fax: (86) 431-87980131.

Sincerely Yours,

Zhanjun Li

18th Jun 2024

Dear Prof. Li,

Thank you for the resubmission of your manuscript to EMBO Molecular. I am pleased to inform you that we will be able to accept your manuscript pending the following final amendments:

- 1) Please address all referee's concerns and implement all his/her suggestions.
- 2) Please pay particular attention to the grammar and syntax and consider running the article by a native English speaker.
- 3) Authors:
 - We note that you currently have together with you, a total of 3 co-corresponding authors. Is that correct? Do you confirm equal contribution of these 3 people, able to take full responsibility for the paper and its content? While there is no limit per se to the number of co-corresponding authors, 3 is rare, and may not reflect as intended to the community.
 - E-mail correspondence to Ruonan Liu could not be delivered. Please update the e-mail address of this author.
- 4) In the main manuscript file, please do the following:
 - Please address all comments suggested by our data editors listed below:
 - o Figure legends:
 1. Please note that the figure legend for figure EV 3f is mislabeled as figure EV 3c in the data information section. This needs to be rectified.
 2. Please note that the exact p values are not provided in the legends of figures 1g-k; 2b-d, f-g; 3c, e-h; 4b-g; 5d-g; 6b, d-i; EV 1i; EV 2d-e; EV 3a-b, f-g.
 3. Please indicate the statistical test used for data analysis in the legend of figure 5b.
 4. Please note that in figures 1g-k; 2b-d, f-g; 3c, e-h; 4b-g; EV 1i; EV 2d-e; EV 3a-b, f-g; there is a mismatch between the annotated p values in the figure legend and the annotated p values in the figure file that should be corrected.
 5. Please note that the error bars are not defined in the legend of figure 1c.
 - Add callouts for Figures 1H, 3E, 4G, and 6D.
 - In Methods, please provide all relevant information about all antibodies and dilutions that were used for each antibody.
 - Indicate in the figure legends the number and nature of replicates and exact p= values, not a range, along with the statistical test used. To keep the figures "clear" some authors found providing an Appendix table Sx with all exact p-values preferable. You are welcome to do this if you want to.
 - Thank you for including Reagents and Tools Table. We would encourage you to use 'Structured Methods', our new Methods format. According to this format, the Methods section should include a Reagents and Tools Table (listing key reagents, experimental models, software and relevant equipment and including their sources and relevant identifiers) followed by a Methods and Protocols section in which we encourage the authors to describe their methods using a step-by-step protocol format with bullet points, to facilitate the adoption of the methodologies across labs. More information on how to adhere to this format as well as downloadable templates (.docx) for the Reagents and Tools Table can be found in our author guidelines: <https://www.embopress.org/page/journal/17574684/authorguide#structuredmethods>
An example of a Method paper with Structured Methods can be found here: <https://www.embopress.org/doi/full/10.1038/s44320-024-00037-6#sec-4>
- Correct the reference citation in the reference list. Where there are more than 10 authors on a paper, 10 will be listed, followed by "et al.". Also, please remove DOIs. Please check "Author Guidelines" for more information.
<https://www.embopress.org/page/journal/17574684/authorguide#referencesformat>
- 5) Movies: Please rename them to Movie EV1 etc. (also in the main text) and zip each movie file with the corresponding movie legend.
- 6) Source Data: Please label all raw images deposited in Biolineage Archive with the corresponding Figure and panel.
- 7) Synopsis:
 - Please check your synopsis text and image before submission with your revised manuscript. Please be aware that in the proof stage minor corrections only are allowed (e.g., typos).
- 8) For more information: This space should be used to list relevant web links for further consultation by our readers. Could you identify some relevant ones and provide such information as well? Some examples are patient associations, relevant databases, OMIM/proteins/genes links, author's websites, etc...
- 9) As part of the EMBO Publications transparent editorial process initiative (see our Editorial at <http://embomolmed.embopress.org/content/2/9/329>), EMBO Molecular Medicine will publish online a Review Process File (RPF) to accompany accepted manuscripts. This file will be published in conjunction with your paper and will include the anonymous referee reports, your point-by-point response and all pertinent correspondence relating to the manuscript. Let us know whether you agree with the publication of the RPF and as here, if you want to remove or not any figures from it prior to publication. Please note that the Authors checklist will be published at the end of the RPF.
- 10) Please provide a point-by-point letter INCLUDING my comments as well as the reviewer's reports and your detailed responses (as Word file).

I look forward to reading a new revised version of your manuscript as soon as possible.

Yours sincerely,

Zeljko Durdevic

*** Instructions to submit your revised manuscript ***

- 1) a .docx formatted version of the manuscript text (including Figure legends and tables)
- 2) Separate figure files*
- 3) supplemental information as Expanded View and/or Appendix. Please carefully check the authors guidelines for formatting Expanded view and Appendix figures and tables at <https://www.embopress.org/page/journal/17574684/authorguide#expandedview>
- 4) a letter INCLUDING the reviewer's reports and your detailed responses to their comments (as Word file).
- 5) The paper explained: EMBO Molecular Medicine articles are accompanied by a summary of the articles to emphasize the major findings in the paper and their medical implications for the non-specialist reader. Please provide a draft summary of your article highlighting
 - the medical issue you are addressing,
 - the results obtained and
 - their clinical impact.This may be edited to ensure that readers understand the significance and context of the research. Please refer to any of our published articles for an example.
- 6) For more information: There is space at the end of each article to list relevant web links for further consultation by our readers. Could you identify some relevant ones and provide such information as well? Some examples are patient associations, relevant databases, OMIM/proteins/genes links, author's websites, etc...
- 7) Author contributions: the contribution of every author must be detailed in a separate section.
- 8) EMBO Molecular Medicine now requires a complete author checklist (<https://www.embopress.org/page/journal/17574684/authorguide>) to be submitted with all revised manuscripts. Please use the checklist as guideline for the sort of information we need WITHIN the manuscript. The checklist should only be filled with page numbers where the information can be found. This is particularly important for animal reporting, antibody dilutions (missing) and exact values and n that should be indicated instead of a range.
- 9) Every published paper now includes a 'Synopsis' to further enhance discoverability. Synopses are displayed on the journal webpage and are freely accessible to all readers. They include a short stand first (maximum of 300 characters, including space)

as well as 2-5 one sentence bullet points that summarise the paper. Please write the bullet points to summarise the key NEW findings. They should be designed to be complementary to the abstract - i.e. not repeat the same text. We encourage inclusion of key acronyms and quantitative information (maximum of 30 words / bullet point). Please use the passive voice. Please attach these in a separate file or send them by email, we will incorporate them accordingly.

You are also welcome to suggest a striking image or visual abstract to illustrate your article. If you do please provide a jpeg file 550 px-wide x 300-800px high.

10) A Conflict of Interest statement should be provided in the main text

11) Please note that we now mandate that all corresponding authors list an ORCID digital identifier. This takes <90 seconds to complete. We encourage all authors to supply an ORCID identifier, which will be linked to their name for unambiguous name identification.

Currently, our records indicate that the ORCID for your account is 0000-0001-6914-8589.

Link Not Available

Photos 400-800 DPI

*Additional important information regarding figures and illustrations can be found at

<https://bit.ly/EMBOPressFigurePreparationGuideline>. See also figure legend preparation guidelines:

<https://www.embopress.org/page/journal/17574684/authorguide#figureformat>

***** Reviewer's comments *****

Referee #2 (Remarks for Author):

The authors provided significant revisions to the original manuscript submission and adequately addressed the majority of my concerns all except for my last critique (#14) (shown below).

14.) Results from Figure 5B and within the figure legend and text descriptions should be referred to as change in relation to differential binding or fold change in abundance of binding and not to upregulated or downregulated.

The inclusion of the new text: "18 bound enhanced proteins" or "0 bound weakened proteins" does not adequately address my critique. I would suggest the following revision "represent 18 bound proteins with enhanced binding affinity" and "represent 0 bound proteins with weakened binding affinity".

Similar to my last review, the authors provide a significant advance to the field in developing a novel rabbit model of cerebral palsy (CP) and providing important mechanistic insight underlying the RhoB p.S73F in causing CP pathology. However, with the inclusion of new text to the revised draft, there are multiple areas in the current draft that need significant editing due to improper grammar, word choice, and overall coherence. I have provided the following examples below that would need significant revision before the current draft is suitable for acceptance to EMBO Molecular Medicine.

1.) New text added in lines 71-75 is not written for clarity. The text "both rabbits and humans are perinatal brain developers" is nonsensical and should be rewritten.

2.) Line 84: please delete "in clinical" to improve clarity.

3.) Line 89: "embryonic efficiency validation" is not common technical nomenclature. Please define this terminology and rewrite for clarity.

4.) Lines 98-102 is not written for clarity. Please edit text to improve clarity.

5.) Lines 124-126 is not written for clarity. Please edit text to improve clarity.

6.) Lines 180-181: "Therefore, we hypnosis the most significant differentially bound protein" should read "Therefore, we hypothesize the most significant differentially bound proteins, ACAT (acetyl-CoA acetyltransferase 1) and LYN (Src family tyrosine kinase), "

7.) Line 193: insert "the" between by and RhoB p.

8.) Lines 200-215 is not written for clarity. Please edit text to improve clarity.

- 9.) Lines 223-233 is not written for clarity. Please edit text to improve clarity.
- 10.) Line 253: the word hypothesised should be spelled hypothesized

To Reviewer #2:

14.) Results from Figure 5B and within the figure legend and text descriptions should be referred to as change in relation to differential binding or fold change in abundance of binding and not to upregulated or downregulated.

The inclusion of the new text: "18 bound enhanced proteins" or "0 bound weakened proteins" does not adequately address my critique. I would suggest the following revision "represent 18 bound proteins with enhanced binding affinity" and "represent 0 bound proteins with weakened binding affinity".

Response: Thank you for your kind suggestion. The description has been changed in line 769 of the revised manuscript.

1.* New text added in lines 71-75 is not written for clarity. The text "both rabbits and humans are perinatal brain developers" is nonsensical and should be rewritten.

Response: Thank you for your kind suggestion. The sentence has now been rewritten for clarity in line 69 of the revised manuscript.

2.* Line 84: please delete "in clinical" to improve clarity.

Response: Thank you for your kind suggestion. The description has been changed in line 85 of the revised manuscript.

3.* Line 89: "embryonic efficiency validation" is not common technical nomenclature. Please define this terminology and rewrite for clarity.

Response: Thank you for your good suggestion. The text has been rewritten in line 90 of the revised manuscript.

4.* Lines 98-102 is not written for clarity. Please edit text to improve clarity.

Response: Thank you for the helpful comment. We have modified the text to improve clarity in line 100 of the revised manuscript.

5.* Lines 124-126 is not written for clarity. Please edit text to improve clarity.

Response: Thank you for your kind suggestion. The description has been changed in line 126 of the revised manuscript.

6.* Lines 180-181: "Therefore, we hypnosis the most significant differentially bound

protein" should read "Therefore, we hypothesize the most significant differentially
bound proteins, ACAT (acetyl-CoA acetyltransferase 1) and LYN (Src family tyrosine
kinase), "

**Response:** Thank you for your valuable suggestion. The description has been changed
in line 185 of the revised manuscript.

**7.*** Line 193: insert "the" between by and RhoB p.

**Response:** Thank you for your valuable feedback. The description has been changed in
line 198 of the revised manuscript.

**8.*** Lines 200-215 is not written for clarity. Please edit text to improve clarity.

**Response:** Thank you for your good suggestion. We have modified the text to improve
clarity in line 200 of the revised manuscript.

**9.*** Lines 223-233 is not written for clarity. Please edit text to improve clarity.

**Response:** Thank you for your valuable suggestion. The sentence has now been re-
written for clarity in line 230 of the revised manuscript.

**10.*** Line 253: the word hypothesised should be spelled hypothesized.

**Response:** Thank you for your insightful feedback. The description has been changed
in line 257 of the revised manuscript.

15th Jul 2024

Dear Prof. Li,

We are pleased to inform you that your manuscript is accepted for publication and is now being sent to our publisher to be included in the next available issue of EMBO Molecular Medicine.
